

# Effects of the interaction of ocean acidification, solar radiation, and warming on biogenic dimethylated sulfur compounds cycling in the Changjiang River Estuary

Shan Jian[1], Jing Zhang[1, 3], Hong-Hai Zhang[1, 3], and Gui-Peng Yang[1, 2, 3]

[1]Key Laboratory of Marine Chemistry Theory and Technology, Ocean University of China, Ministry of Education, Qingdao 266100, China

[2]Laboratory for Marine Ecology and Environmental Science, Qingdao National Laboratory for Marine Science and Technology, Qingdao 266071, China

[3]Institute of Marine Chemistry, Ocean University of China, Qingdao 266100, China

*Correspondence to*: Gui-Peng Yang (gpyang@ouc.edu.cn)

**Abstract.** Ocean acidification (OA) affects marine primary productivity and community structure, and therefore may influence the biogeochemical cycles of volatile biogenic dimethyl sulfide (DMS) and its precursor dimethylsulfoniopropionate (DMSP) and photochemical oxidation product dimethyl sulfoxide (DMSO). A 23-day incubation experiment on board was conducted to investigate the short-term response of biogenic sulfur compounds

production and cycling to OA in the Changjiang River Estuary and further understand its effects on biogenic sulfur compounds. Result showed that phytoplankton abundance and species presented remarkable differences under three different pH levels in the late stage of the experiment. A significant reduction in chlorophyll *a* (Chl-*a*), DMS, particulate DMSP (DMSPp), and dissolved DMSO (DMSOd) concentrations was identified under high $CO_2$ levels. Moreover, minimal change was observed in the production of dissolved DMSP (DMSPd) and particulate DMSO (DMSOp) among treatments. The

ratios of DMS, total DMSP (DMSPt), and total DMSO (DMSOt) to Chl-*a* were also not affected by a change in pH. In addition, DMS and DMSOd were highly related to mean bacterial abundance under three pH levels. Additional incubation experiments on light and temperature showed that the influence of pH on productions of dimethylated sulfur compounds also depended on solar radiation and temperature conditions. DMS photodegradation rate increased with decreasing pH under full-spectrum natural light and UVB light. Thus, OA may lead to decreasing DMS concentrations in the surface seawater.

Light and temperature conditions also play an important role in the production and cycling of biogenic sulfur compounds.

Keywords: Dimethylated sulfur compounds, Ocean acidification, Solar radiation, Warming, Bacteria, Phytoplankton

## 1 Introduction

The current atmospheric $CO_2$ concentration ($p$CO$_2$) is around 380 µatm and rising at an unprecedented rate. According to the Special Report on Emissions Scenarios by the Intergovernmental Panel on Climate Change, $p$CO$_2$ is predicted to reach

approximately 1,000 µatm by 2100 (Nakicenovic and Swart, 2000), thereby decreasing the surface ocean pH levels by 0.2–



0.5 units (Caldeira and Wickett, 2003). The primary consequence of the inexorable increase in oceans' acidity is a change in seawater carbonate system and a decrease in the saturation state of calcite and aragonite carbonate. This change could affect marine ecological environment and primary productivity, as more $CO_2$ (low pH) will affect physiological process of marine organism (Fu et al., 2007; Măancon et al., 2016; Orr et al., 2005). The oceans are an important source of some atmosphere

trace gases which affect atmospheric chemistry and global climate (Arnold et al., 2013; Asher et al., 2016; Park et al., 2014).

Dimethyl sulfide (DMS) is an important climatically active biogenic gas. It is produced by enzymatic cleavage of dimethylsulfoniopropionate (DMSP) (Gabric et al., 2010), which is synthesized by marine phytoplankton as a phytoplankton-derived precursor of DMS. DMS is the dominant volatile sulfur compound in ocean surface waters (Quinn and Bates, 2011) and its emissions from the ocean to the atmosphere corresponds to almost 90% of the marine biogenic

sulfur in the atmosphere (Wakeham, 1986). Once emitted to atmosphere, DMS undergoes rapid oxidation to produce particles which, through direct and indirect interactions with incoming solar radiation, affect planetary albedo (Barnes et al., 2006; Charlson et al., 1987; Rap et al., 2013). Shaw hypothesized that DMS and sulfate aerosols are linked to global climate. This link was further elaborated by Charlson et al. (1987). Consequently, ocean acidification (OA)-induced changes in the primary productivity might impact on the production rate and sea-to-air emission of DMS and these impacts might further

affect cloud formation and climate.

DMSP, as the main precursor of DMS, is produced by various phytoplankton species and presents an important effect on the biogeochemical cycle of climatically active trace gas DMS (Keller et al., 1989). DMSP is released from phytoplankton cells into the dissolved phase through active exudation or when cells are lysed during grazing (Wolfe et al., 2000), viral attack (Malin et al., 1998), or senescence (Stefels et al., 2007). The conversion of DMSP to DMS is controlled by a number

of chemical and biological processes (Archer et al., 2002). DMSP is degraded in seawater via two main pathways. DMSP-lyase pathway contributes only a considerably small fraction to DMSP metabolism in seawater. The most preferred and dominant process is the dimethylation/demethiolation pathway, which diverts sulfur away from DMS production (Kiene and Bates, 1990). However, whether shifts in DMS yield occur under OA condition is unknown.

Dimethyl sulfoxide (DMSO) is the major nonvolatile dimethyl sulfur pool in seawater and also plays an important role in

biogeochemical cycle of DMS. Marine DMSO is derived mainly from DMS through abiotic photolysis and biological consumption (Brimblecombe and Shooter, 1986; Kieber et al., 1996; Toole and Siegel, 2004). DMSO was initially conceived as a sink for DMS. Nevertheless, DMSO was later found a potential source of DMS (Hatton et al., 2012). DMSO can be biologically reduced to DMS via enzymatic reactions that might depend on reductases (Spiese et al., 2009). Thus, DMSO is a key compound in the complex redox loop that is involved in marine sulfur cycle because it can be both an end

product of DMS oxidation and a precursor of DMS, which potentially plays an important role in climate regulation (Charlson et al., 1987; Deschaseaux et al., 2014).

Along with ocean acidification, global warming associated with the increasing atmospheric $CO_2$ accumulation is leading to a greenhouse ocean, which is characterized by increased sea surface temperature and a shoaling of the upper mixed layer (Capotondi et al., 2012; Doney, 2006; Gao et al., 2012a; Hays et al., 2005). The declining stratospheric ozone over the globe





and change in the upper mixed layer would enhance the levels of ultraviolet radiation (UVR) (280 to 400 nm) reaching the sea surface. The ocean undergoes multiple environmental changes. Other climatic ecological stressor or factors would probably alter the effects of ocean acidification on the production and consumption process of DMS in both direct and indirect ways (Arnold et al., 2013). A recent study predicts that pH level can influence the photodegradation of DMS (Bouillon and Miller, 2005). The photochemical process of DMS in the surface water would change due to the changing light level and seawater pH level. So these concomitant global change variables should be taken into consideration to better explain the effect of acidification on DMS.

The effects of simulated future $CO_2$-induced seawater acidification on DMS production are still controversial (Avgoustidi et al., 2012; Kerrison et al., 2012; Kim et al., 2010; Vogt et al., 2008). Hence, we conducted a shipboard incubation experiment in the Changjiang River Estuary to further investigate the OA effects in the cycling of marine DMS, DMSP, and DMSO under three pH treatments. Moreover, we examined the photolysis rate and concentration changes of DMS under the dual stressors of changing pH and solar radiation/temperature and assessed the coupling effects of OA, solar radiation and warming on biogenic dimethylated sulfur compounds cycling.

## 2 Materials and methods

### 2.1 Experimental design

Deck incubation experiment of ocean acidification was conducted aboard the R/V 'Run Jiang No. 1' in March 7–23, 2016 and continued for 5 days at the laboratory. The surface seawater (5 m) sample, encompassing a range of environmental conditions, was collected from the station A6-9 (123.50 ° E, 30.56 ° N) in the Changjiang River Estuary using 20 L Niskin bottles mounted on a CTD rosette. Environmental conditions of the sampling station were measured by CTD. Initial temperature and salinity were 12.33 ℃ and 33.79‰, respectively. Phytoplankton communities were grown in nine 20 L barrels (polypropylene, Nalgene) that were labeled M1, M2, M3, M4, M5, M6 M7, M8, and M9. $CO_2$-saturated and filtered (0.2 μm of polyethersulfone membranes) seawater was gradually added to the nine incubation barrels until the target pH levels were reached on day 1. The incubation experiment was initially equilibrated to the following conditions: (1) three were at ambient levels of pH (8.1, M1–M3, LC group) without any subsequent processing step; (2) three were at medium levels and expected at the end of this century (7.9, M4–M6, MC group), and (3) three were at relatively low pH levels as predicted for the middle of the next century (7.7, M7–M9, HC group). All treatments were exposed to sunlight in a flow-through water bath to maintain treatments at their in situ temperature. Acidification incubation samples were gathered at 9:00 am daily or every second day. Additionally, three kinds of sulfur compounds (DMS, DMSP, and DMSO) and chlorophyll *a* (Chl-*a*) samples were collected for analysis from day 1 to day 22. Seawater pH was continually measured using a pH meter (S210 SevenCompact™, Mettler, Germany) and the precision is ± 0.002.

Incubations of filtered surface seawater with same DMS concentrations (5 nmol L$^{-1}$) were distributed into 150 mL quartz tubes and subsequently allocated into different spectral treatments to examine the combined effects of OA and solar radiation;



the treatments were as follows: (1) quartz tubes (full-spectrum natural light), (2) Mylar-D-wrapped quartz tubes transmitting 60%−80% of UVA and small amount of UVB, (3) UF3 Plexiglas-enclosed quartz tubes attenuating essentially all UVA and UVB with 92% of visible light transmitted, (4) three layers of aluminum foil-wrapped quartz tubes (dark) evaluating the dark loss of DMS, and (5) UVB radiation (results for full-spectrum natural light minus those for UVB-filtered light) (Hatton,

2002). All samples were exposed to ambient solar radiation for 8 h and sampled every 2 h. Temperature experiments were also carried out with unfiltered water in quartz bottle under in situ temperature at 12 ℃ and high temperature at 18 ℃ for 8 h. The temperature was continuously controlled by circulating in situ seawater, hot water, and ice. The pH levels for solar radiation and temperature experiments were controlled with the same methods as previously described.

## 2.2 Analytical procedures

Triplicate DMS samples were measured immediately after collection with a cryogenic purge-and-trap preconcentration technique in accordance with the methods described by Zhang et al. (2008). In brief, 2 mL aliquot of the sample was collected into a glass bubbling chamber, stripped with high-purity nitrogen, trapped in a 1/16 Teflon tube, and immersed in liquid nitrogen. The trapped gases were desorbed with hot water (> 90 ℃) and then introduced into a gas chromatograph (GC, Shimadzu GC-2014) with a flame photometric detector for analysis. A 3 m × 3 mm glass column packed with 10%

DGES on Chromosorb W-AW-DMCS was used to separate sulfur gases at 70 ℃. The precision of the method was better than 5%, and the detection limit was 0.1 pmol of DMS.

The samples for dissolved DMSP (DMSPd) were separated by gravity filtering with a Whatman GF/F filter. For DMSPd samples, 4 mL of sample was transferred to a vial containing 40 μL of concentrated sulfuric acid and then sealed. For the total DMSP (DMSPt) samples, 100 μL of 50% sulfuric acid was directly added to 10 mL of unfiltered seawater samples and

then sealed. Before measurement, 300 μL of 10 mol L$^{-1}$ KOH was injected into 2 mL of preservative DMSP sample and incubated in the dark at 4 ℃ for at least 24 h, thereby allowing the complete conversion of DMSP into DMS. Liberated DMS samples were determined using the above-mentioned methods.

The gravity-filtered samples for dissolved DMSO (DMSOd) were transferred to an immediately sealed vial that contained 20 μL of 25% hydrochloric acid. Approximately 50 μL of 25% hydrochloric acid was directly added to 10 mL of unfiltered

seawater for the total DMSO (DMSOt) samples for analysis. Prior to the analysis, the samples were purged with ultrapure N$_2$ for 10 min at a flow rate of 100 mL min$^{-1}$ to remove initial DMS interferences. DMSO samples were analyzed by DMS reduction with titanium trichloride at 55 ℃ in the dark for 1 h, in accordance with the procedures described by Kiene and Gerard (1994). The reduction of DMSO with TiCl3 can be expressed by the following reaction:

$$C_2H_6SO + 2TiCl_3 + 2HCl \rightarrow C_2H_6S + 2Ti(s) + 4 Cl_2(g) + H_2O$$

Chl-$a$ concentrations were measured with a Hitachi F-4500 fluorescence spectrophotometer after filtration of 300 mL of seawater sample with a Whatman GF/F glass fiber filter; subsequently, the sample was extracted in 90% acetone according to the methods described by Parsons et al. (1984).



Approximately 200 mL of subsamples for phytoplankton analysis were fixed in neutralized formalin to a final concentration of 4% and concentrated to 10 mL by sedimentation. Phytoplankton species were identified and counted by microscopic examination under an inverted microscope (LEICA DRB) at magnifications of 400–600×.

Total bacterial abundances were measured using a LEICA DM5000B fluorescence spectrophotometer. Briefly, 20 mL of seawater samples were preserved with formaldehyde (3.4% final concentration) and kept in sterile 50 mL screw-cap plastic bottles in the dark at 4 ℃. To identify the bacteria, the samples were stained with 4',6-diamidino-2-phenylindole (a highly specific and sensitive fluorescing DNA stain) and counted by fluorescence microscopy according to the methods described by Porter and Feig (1980).

Total dissolved inorganic carbon (DIC) was determined by a DIC analyzer (AS-C2, Apollo SciTech Inc., Georgia, USA). A sample of 0.5 mL was acidified by 0.5 mL 10% phosphoric acid and then the extracted $CO_2$ gas was measured using a nondispersive infrared (NDIR) $CO_2$ detector (LI-6262, Li-COR Inc., USA) with a precision of 0.1%. The total alkalinity (TA) and $p$CO$_2$ in sea water were calculated from DIC and pH using the CO2SYS (as refitted by Dickson and Millero (1987)).

### 2.3 Statistical analyses

Statistical analysis was performed using SPSS version 18.0 (SPSS Inc., IBM, USA). Pearson's correlation coefficient and probability (p) were calculated to evaluate the quality of the fit when variables were normally distributed. T-test was used to determine whether a significant difference existed between two treatments. Variability in the concentrations of biogenic dimethylated sulfur compounds was analyzed using two-way ANOVA with pH and temperature/light as fixed factors and concentration as a random factor to understand whether an interaction existed between pH and temperature/light on concentration. Value at $p \leq 0.05$ was considered statistically significant. Linear correlation analyses were used to determine the response of DMS, DMSO concentrations, and bacterial abundance to OA using Origin 9.1.

### 3 Results

### 3.1 The effect of OA on Chl-*a*, DMS, DMSP and DMSO concentrations, and bacterial abundance

### 3.1.1 Variation in pH

At the start of the experiment, the seawater pH values in the incubation barrels were adjusted to the desired pH levels and the initial carbonate parameters are shown in Table 1. Seawater pH was adjusted with $CO_2$-saturated seawater during the experiment to maintain a stable pH environment. In the first week, pH level was comparatively stable with a better control, but pH fluctuated obviously and was difficult to control in the stable and decline phases of algal growth. As shown in Fig. 1, the pH showed relatively apparent fluctuations on days 9-12 and days 18-20. This might be affected by multiple biological activities, in which the effect of microbial respiration should be the main influencing factor. $CO_2$ emitted from microbial



respiration can influence the carbonate balance system in seawater. Overall, the pH was controlled at relatively stable levels in the three treatments.

### 3.1.2 Phytoplankton biomass and Chl-*a*

Phytoplankton from station A6-9 included diatom, dinoflagellate, chrysophyceae, and cryptophyceae with a mean abundance of 2135 cells $L^{-1}$. Diatoms and dinoflagellates dominated the phytoplankton community, and their proportions were 67% and 32%, respectively. The seawater samples of this station initially enclosed within incubation barrels exhibited a low-nutrient environment (0.216 μmol $L^{-1}$ $NO_3^-$, 0.0304 μmol $L^{-1}$ $NO_2^-$, 0.963 μmol $L^{-1}$ $NH_4^+$, 0.706 μmol $L^{-1}$ $PO_4^{3-}$, and 0.767 μmol $L^{-1}$ $SiO_3^{2-}$) and low Chl-*a* concentrations (0.460 μg $L^{-1}$). Thus, in order to guarantee that phytoplankton biomass could be determined during incubations, nutrients (17 μmol $L^{-1}$ $NO_3^-$ and 1.0 μmol $L^{-1}$ $PO_4^{3-}$) were added to each incubation barrel on day 1. The response of dimethylated sulfur compounds was assessed during incubations of natural seawater assemblages under three triplicated $CO_2$ treatments. The mean dimethylated sulfur compounds and Chl-*a* concentrations and the bacterial abundance recorded in different treatments are shown in Table 2.

At the late stage of the experiment, significant differences in phytoplankton biomass were observed among the three treatments. Phytoplankton biomass and species were less in HC than in LC, and chrysophyceae (e.g., coccolith) and cryptophyceae (e.g., *Cryptomonas* sp. and *Rhodomonas* sp.) disappeared during HC treatment. This result revealed that primary productivity was influenced by varied levels of acidity and the pH levels could influence the phytoplankton community structure. The results can be explained using the conclusion obtained by Deppeler et al. (2017) who supported the identification of a tipping point in the marine microbial community response to $CO_2$ between 953 μatm and 1140 μatm. When phytoplankton was exposed to a low pH level beyond their limits of tolerance, their growth rates and productivity were declined obviously. During the experiment, diatom became increasingly dominant in the phytoplankton biomass. The average abundance of diatom was 4360 cells $L^{-1}$ in MC treatment, which was higher than that in LC (2710 cells $L^{-1}$) and HC (980 cells $L^{-1}$) treatments on day 18. However, the largest dinoflagellate abundance was observed in HC treatment with a value of 2210 cells $L^{-1}$ on day 18. This shows that the effects of different acidity conditions on phytoplankton are different. Appropriate decrease in seawater pH may promote the growth of diatom and excessive reduction of acidity would retard its growth. Compared with diatom, low pH conditions are more favorable for dinoflagellate growth.

Chl-*a* is the main indicator of phytoplankton biomass, and its concentrations during the experiment were associated with phytoplankton biomass. The change in Chl-*a* concentrations for each of the nine barrels and averages over the triplicate treatments is shown in Fig. 2. The Chl-*a* concentrations of three parallel treatments which had the same pH level showed similar variation trends and the curves of Chl-*a* concentration include logarithmic phase, stable and decline phases, indicating that the communities of each paratactic treatment under different acidic conditions showed similar behaviors over the course of the experiment. Following nutrient addition, a strong phytoplankton bloom proceeded in all treatments. Phytoplankton abundances (Chl-*a*) peaked at nearly the same time (days 7–8) in all of the barrels. The maximum Chl-*a* concentrations of 12.6 μg $L^{-1}$ appeared in M3 (LC treatment) and the lowest concentrations of 4.27 μg $L^{-1}$ was observed in



M9 (HC treatment) (Fig. 2a). Subsequent to the remarkable Chl-*a* peak, the concentrations rapidly decreased, and the minimum value was observed at the end of the experiment. The mean values of Chl-*a* concentrations for the whole experiment were 3.14 ±3.22, 2.60 ±2.31, and 2.14 ±1.72 μg L$^{-1}$ for LC, MC, and HC treatments, respectively (Fig. 2b). As can be seen, a significant decrease in mean Chl-*a* concentrations under MC and HC treatments was observed, with a mean 17%

and 32% decrease, respectively. However, these large decrease showed no significant statistical differences between the treatments compared with the LC treatments during the entire experiment (for MC: T = 0.53, DF = 28, p = 0.600; for HC: T = 1.055, DF = 28, p = 0.300). Nevertheless, obvious differences were observed in the Chl-*a* concentrations between the LC and HC treatments during days 8–13 (T-test: T = 2.393, DF = 8, p = 0.044), indicating that change of seawater pH had a great influence on phytoplankton biomass in the intermediate stage of this acidification experiment.

### 3.1.3 Sulfur compounds

DMS production followed the development and decline of the phytoplankton bloom (Figs. 2 and 3). Maximum DMS concentrations coincided with the highest Chl-*a* concentrations in LC (55.76 nmol L$^{-1}$) and MC (44.86 nmol L$^{-1}$) treatments. However, the maximum DMS concentrations in the HC treatment (30.58 nmol L$^{-1}$) were delayed by 2 days relative to the highest Chl-*a* concentrations. This may be because the acidification condition has a greater impact on the production and

release of DMS during the logarithmic growth phase of algae. The rate of increase was significantly higher in LC treatments than that in MC and HC treatments. The time integrated mean concentration of DMS at HC (10.65 nmol L$^{-1}$) and MC (13.42 nmol L$^{-1}$) levels was 35% and 19% lower than that at LC level (16.50 nmol L$^{-1}$), respectively. DMS showed statistically significant differences between LC and HC treatments during days 6–11 (T = 2.492, DF = 8, p = 0.037). Results were consistent with those of Wingenter et al. (2007).

For all $CO_2$ treatments, DMSPd and particulate DMSP (DMSPp) concentrations ranged within 2.88–83.21 nmol L$^{-1}$ (mean 19.37 nmol L$^{-1}$) and 12.91–147.63 nmol L$^{-1}$ (mean 60.65 nmol L$^{-1}$) in the incubation barrels, respectively (Fig. 4 and Table 2). The DMSPd concentrations in M1, M3, M4, M8, and M9 increased continuously until day 8 and decreased afterward until the end of the experiment. M2, M5, M6, and M7 also exhibited a secondary increase during the final 4–5 days of the experiment. The maximum of M7 (HC treatment) was delayed by 3 days (Fig. 4a). The pH decrease from 7.9 to

7.7 caused an increase in the mean DMSPd from 23.7 nmol L$^{-1}$ in the MC treatment to 27.7 nmol L$^{-1}$ in the HC treatment (Fig. 4b) from day 6 to 16 of the experiment. However, in general, no difference was observed in DMSPd concentrations among treatments. DMSPp concentrations varied over the time course of the experiment. Although mean concentrations decreased by 17% and 21% under medium and high $CO_2$ treatments, respectively, these differences were also not statistically significant. The effects on DMSP were rather insignificant than those on DMS, it seems that there was a great influence of

acidification on the conversion of DMSP to DMS.

Variations in DMSOd and DMSOp concentrations in all cultures are shown in Figs. 5a and 5c. The temporal trends of DMSOd in the nine treatments were significantly different. Under LC treatments, DMSOd concentrations peaked on days 8, 9, and 18 (Figs. 5a and 5b). By contrast, the DMSOd concentrations of MC and HC treatments, particularly M5, M6, M8,





and M9, showed a slight increase, which presented no concurrence with the trend in Chl-*a* or DMS (Figs. 2a, 3a, and 5a). In all treatments, an overall increase was observed in DMSOp concentrations with time. The DMSOp concentrations in most $CO_2$ treatments rapidly increased during the initial part of the experiment with a maximum value at days 10–11 and remained stable afterward (Figs. 5c and 5d). DMSOd concentrations showed notable difference during days 3-10 and $CO_2$ treatment

exerted no significant effects on DMSOp concentrations during the experiment.

### 3.1.4 Bacteria

Bacteria can utilize DMSP as a growth substrate to produce DMS (Toole et al., 2008; Yoch et al., 1997) and they can convert DMS and DMSO into each other through oxidation or reduction processes, thereby playing an essential role in the biogeochemical cycle of sulfur. Piontek et al. (2010) showed that bacterial ectoenzyme activities are sensitive to changes in

pH levels; thus, the present work investigated the potential effect of bacteria on dimethylated sulfur compounds concentrations under different pH conditions. In this research, the mean bacterial abundance increased with increased $CO_2$ levels and ranged from $0.29 \times 10^8$ cells $L^{-1}$ to $9.60 \times 10^8$ cells $L^{-1}$, with the maximum appearing in HC treatment. This result is consistent with that obtained by Deppeler et al. (2017) who suggested that bacterial abundance in treatment $\geq 634$ μatm increased with increasing $CO_2$. However, examination of individual coefficients revealed that these differences under pH

treatments were not statistically significant. Previous literatures (Grossart et al., 2006; Allgaier et al., 2008; Liu et al., 2010; Avgoustidi et al., 2012) indicated that high $CO_2$ levels had little or no effect on bacterial abundance. The difference of bacterial abundance among treatments would not be directly caused by pH change. Bacteria appeared to be relatively tolerant to ocean acidification, and bacterial abundance was indirectly affected by the responses of phytoplankton to ocean acidification (Grossart et al., 2006; Piontek et al., 2013; Deppeler et al., 2017). Thus, the high bacterial abundance in M7-M9

group may result from an increased dead phytoplankton at reduced pH. Algal-derived organic matter was released during this decline process of phytoplankton and provided a labile energy and carbon source to bacteria in form of structural cell components and storage glucan (Smith et al., 1995).

### 3.2 DMS and DMSP concentration response to the interaction among OA, solar radiation, and warming

### 3.2.1 Solar radiation

In order to assess the community-level response to the ocean change in future $CO_2$ and light conditions, photolysis rate constants ($K$, $d^{-1}$) for DMS were calculated. The result showed that the natural logarithm of DMS concentrations had a good linear relationship with time, so the photodegradation of DMS follows a pseudo-first-order kinetics (Bouillon and Miller, 2005; Brugger et al., 1998). $K$ can be calculated by the slope of the regression line of natural logarithm of DMS concentrations against time. DMS loss was observed under full-spectrum natural light, UVB, UVA, visible light, and dark

conditions in each pH treatment. DMS photolysis rate constants were significantly different in various conditions (Table 3). Furthermore, no significant loss in DMS was observed in the dark during control experiments. The rate constants of DMS



photolysis were in the range of 4.02–6.32, 1.13–4.73, 1.35–2.37, and 0.24–0.52 d$^{-1}$ under full-spectrum natural light, UVB, UVA, and visible light with average values of 5.18, 3.02, 1.75, and 0.41 d$^{-1}$, respectively (Table 3). The rate constants of DMS photolysis under full-spectrum natural light were higher than those under UVB, UVA, and visible light. The UVB made an important contribution to DMS photodegradation. The rate constants of DMS photolysis increased with decreasing

pH levels under full-spectrum natural light and UVB. However, a complete and contrasting result was obtained under UVA and visible light. The contributions to the total photolysis of UVB under decreasing pH conditions were increased from 28.0% to 74.9%. On the contrary, the contributions of UVA and visible light were decreased from 59.1% to 21.3% and from 12.9% to 3.83%, respectively (Table 3). These rate constants and the initial DMS concentration were used to calculate the DMS photolysis turnover time (τphoto, d). The turnover time for the upper 1 m of the water column under full-spectrum natural

light ranged from 0.76 d to 1.19 d with an average value of 0.96 d.

### 3.2.2 Temperature

DMS and DMSPp concentrations were recorded after 8 h under six cross-over incubation experiments with a full factorial combination of LC, MC, and HC treatments under ambient temperature and high-temperature treatments (12 ℃ + pH 8.1, 12 ℃ + pH 7.9, 12 ℃ + pH 7.7, 18 ℃ + pH 8.1, 18 ℃ + pH 7.9, and 18 ℃ + pH 7.7; Fig. 6).

The mean DMS ranged from 3.90 ±0.11 nmol L$^{-1}$ (12 ℃ + pH 7.7) to 6.80 ±0.22 nmol L$^{-1}$ (18 ℃ + pH 7.9) (Fig. 6a). Under ambient temperature condition (12 ℃), the average DMS concentration at pH 7.7 (3.90 ±0.11 nmol L$^{-1}$) was about 35% lower than that at pH 8.1 (6.01 ±0.70 nmol L$^{-1}$). No statistically significant difference was observed in the mean DMS concentrations between LC and MC (p = 0.328). However, statistically significant differences were observed between LC and HC (p = 0.005). For DMSPp, the average concentrations at 18 ℃ were higher than that at 12 ℃. For instance, the mean

DMSPp increased from 24.3 ±1.37 nmol L$^{-1}$ at 12 ℃ under pH 8.1 treatment to 27.0 ±1.01 nmol L$^{-1}$ at 18 ℃ under pH 8.1 treatment, 31.6 ±1.28 nmol L$^{-1}$ at 18 ℃ under pH 7.9, and 36.0 ±0.40 nmol L$^{-1}$ at 18 ℃ under pH 7.7 treatment when the temperature increased from 12 ℃ to 18 ℃ (Fig. 6b). Temperature significantly affected (p < 0.001) the DMSPp concentrations, and the effects outweighed the effects of pH decrease. The concentrations of DMSPp showed significant differences among three treatments (p ≤ 0.008) for high-temperature treatments and minimal differences under ambient

temperature treatments. Results from two-way ANOVA illustrated the interaction between temperature and pH on DMSPp concentration (p = 0.001). This indicated that the pH and temperature influenced the biological production of the dimethylated sulfur compounds in seawater.


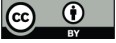

## 4 Discussion

### 4.1 The effect of OA

#### 4.1.1 The effect of OA on the variation of DMS, DMSP, DMSO, and Chl-*a* concentrations

The mean concentration of Chl-*a* was significantly lower under HC treatments with decreases of 32%. The low pH
conditions may influence the membrane electrochemical potential, enzyme activity (Kramer et al., 2003; Milligan et al.,
2009), and $CO_2$-concentrating mechanisms (CCMs) in phytoplankton (Rost et al., 2003; Wu et al., 2010), which affect
primary productivity and growth rate (Gao et al., 2012a, 2012b). The phytoplankton biomass at this station, particularly that
of coccolith, *Cryptomonas* sp. and *Rhodomonas* sp., was negatively affected at the lowest pH level.

Differences in Chl-*a* could be responsible for differences in the production of dimethylated sulfur compounds. The DMS
concentrations measured in the experiment were significantly low under HC treatments (Fig. 3) with mean decreases of 35%.
This result was in agreement with the study of Avgoustidi (2006) that also showed substantially low DMS production under
high $CO_2$ levels. Nevertheless, changes in pH, which was adjusted during the experiment, showed no or minimal effects on
DMSPd/p concentrations. This phenomenon may be caused by the higher percentage of dinoflagellates, which are a prolific
DMSP producer, in HC treatment than that in LC treatment, although the total amount of algae was lower in the former than
in the latter. Similarly, DMSOd/p concentrations presented distinct patterns with DMS and showed less difference among
three treatments, particularly DMSOp. The effects on DMSP and DMSO were less pronounced than that for DMS, it seems
that there was a great influence of acidification on the conversion of DMSP/DMSO to DMS. It could be inferred that
changes in other biological and physical processes in response to changes in pH cannot be ruled out. The processes involved
in DMS conversion included bacterial and phytoplankton production/uptake, viral lysis, and grazing (Wingenter et al., 2007).
Low pH affected one or more of these processes, thereby possibly causing the large decrease in DMS. Thus, differences in
DMS may be seen as a result of changes in the whole ecosystems caused by different pH levels.

#### 4.1.2 Relationships among DMS, DMSP, DMSO, Chl-*a*, and bacteria

A correlation coefficient matrix was calculated based on dimethylated sulfur compounds and Chl-*a* concentrations in the
three different $CO_2$ treatments (Table 4) to evaluate quantitative similarities and differences of mean DMS, DMSP, DMSO,
and Chl-*a* concentrations in different $CO_2$ treatments. Chl-*a* and DMS, Chl-*a* and DMSPd, DMS and DMSPd, and DMSPp
and DMSOp all showed significant positive correlations in all treatments. Moreover, Chl-*a* and DMSPp, DMS and DMSPp,
DMS and DMSOd, and DMSPd and DMSOd showed a correlation in the LC treatments. However, no correlation was
observed in the MC and HC treatments. On the contrary, DMSOp and DMSOd, and DMSPp and DMSOd showed
significant positive correlations only in HC treatments. These relationships implied that OA may affect the biological
activity of algae and the production and consumption processes of sulfur compounds, which consequently affect their
distribution and circulation.





Given that algae are the sole producers of DMSP, direct correlations were observed between both DMS and DMSP levels and Chl-*a*. The correlations were relativity weakened with increased $CO_2$ level. The correlation between DMSPp and Chl-*a* also disappeared under relatively high $CO_2$ levels. This phenomenon might be caused by the changed amount and proportion of various algae under high $CO_2$ conditions. Furthermore, DMS concentrations showed a strong correlation with DMSPd under LC treatment. By contrast, a weak correlation was observed with DMSPp under LC treatment, and no correlation was observed under MC and HC treatments. An overlap between the peaks of DMS and DMSPd and peaks of Chl-*a* is shown in Figs. 2, 3, and 4. DMS and DMSPd also showed their highest peaks with relatively low DMSPp concentrations. This phenomenon might indicate that grazing by zooplankton or senescence of algal cells presents a stronger effect on DMS and DMSP concentrations in MC and HC treatments than in LC treatment. Grazing zooplankton on DMSP-producing algae can increase DMSPd and DMS concentrations (Archer et al., 2002; Simó et al., 2002; Wolfe and Steinke, 1996). Nguyen et al. (1988) showed that DMSP-producing algae are largely responsible for the production of DMSPd and DMS in seawater during their senescence.

A significant correlation between DMS and DMSOd was observed in LC treatments (Table 4). The result was in accordance with the findings of Hatton et al. (2004) and Zindler-Schlundt et al. (2015) who showed that DMSO and DMS are closely related because of the direct formation of DMSO via bacterial DMS oxidation. The close link between DMS and DMSOd was not observed in MC and HC treatments. The changes in the production, consumption, and degradation processes which caused by the decreasing pH might mask the relationship between DMS and DMSOd. In addition, DMSPd concentration showed an obvious correlation with DMSOd only in LC treatments for the entire duration of the experiment (Table 4). DMSPd may be cleaved to DMS by specific bacteria that contain DMSP lyase (Curson et al., 2008), followed by the bacterial oxidation of DMS to DMSOd. This relationship was not found in MC and HC treatments. The phenomenon might be caused by the change in phytoplankton community and bacterial oxidation of DMS to DMSO.

The bacterial consumption of DMS is an important process, and one which if rapid enough, cannot result in increase in seawater DMS concentrations (Kiene and Bates, 1990). The mean bacterial abundance under HC treatments was a maximum value of $3.74 \times 10^8$ cell $L^{-1}$ (with a minimum mean DMS concentration of 10.65 nmol $L^{-1}$) and showed a significant relationship with DMS concentration (Fig. 7). This result indicated that lowered DMS concentrations under HC treatment were in part a result of the enhanced bacterial uptake of DMS in high $CO_2$ conditions. Crawfurd et al. (2016) and Spilling et al. (2016) reported that bacterial grazing and viral lysis were higher in the high $CO_2$ treatments during periods of the experiment, and these processes can accelerate the release of dimethylated sulfur compounds from phytoplankton cells to some extent (Dacey and Wakeham, 1986). This agrees well with our inference mentioned in section 4.1.1. In addition, bacterial abundance was correlated with DMS and DMSOd concentrations in the three kinds of treatments. This result suggested that bacterial degradation of DMSP was an important source of DMS, and bacterial production of DMSOd might be an important source of DMSOd in the incubation experiment.



### 4.1.3 DMS/Chl-*a*, DMSPt/Chl-*a*, and DMSOt/Chl-*a* ratios

The ratios of DMS/Chl-*a*, DMSPt/Chl-*a*, and DMSOt/Chl-*a* are presented in Fig. 8. The ratios provide a good indication of the relative strength of DMS, DMSP, and DMSO production by the phytoplankton assemblage at different stages of the experiment. Moreover, these ratios can highlight where the ecosystem shifts may have occurred. The ratios showed a drastic

decline during the first weekend and subsequently a slow increase in the later period. The ratios of DMS to Chl-*a* showed minimal differences at the three $CO_2$ levels with an average of 8% less DMS produced per Chl-*a* under MC treatment. However, this trend was reversed for the ratios of DMSPt and DMSOt to Chl-*a* separately with 22% and 15% more production per Chl-*a* in the HC treatment compared with the LC treatment. Under MC treatments, the highest ratio values of DMS/Chl-*a*, DMSPt/Chl-*a* and DMSOt/Chl-*a* were observed on the last day. These ratios showed similar trends during the

experiment, and no pronounced difference was found in these ratios at the three $CO_2$ levels. The results indicated no difference in the dimethylated sulfur compounds production in terms of phytoplankton biomass.

### 4.2 The effect of the interaction between OA and environmental factors

The photolysis rate constants of DMS were strongly affected by solar radiation and OA. The net effects of OA on DMS production were largely dependent on light conditions. Relatively, significant change was observed in $K_{UVB}$ and $K_{UVA}$,

particularly $K_{UVB}$ increased by four times at pH 7.7 compared at pH 8.1, indicating that reduced pH levels accelerated the photodegradation of DMS under UVB conditions. The result was mainly because photodegradation of DMS was related to inducement of some oxidants (such as singlet oxygen, hydroxyl radical, hydrogen peroxide, and photoactivated CDOM) (Kieber et al., 1996; Brimblecombe and Shooter, 1986) and some ions (such as $HCO_3^-$, $CO_3^{2-}$, $NO_3^-$, and $Br^-$) (Bouillon and Miller, 2005). The short wavelength UVB played a decisive role in the formation and reaction of these oxidants and pH

simultaneously influenced the free-radical production/scavenging processes involving ions, which may be responsible for the variation in photodegradation rate under different sunlight and pH conditions. Although the interaction of acidification and solar radiation is considerably complex, an overall decrease of pH in UVB would result in an eventual decrease in DMS, indicating that ocean acidification can promote DMS photooxidation under UVB. OA is not proceeding in isolation (Gao et al., 2012). The effect of the interaction between OA and environmental conditions complicates the overall ecosystem

response. Hence, comprehensive consideration of OA and solar radiation can better interpret and understand feedbacks between OA and global climatic change.

Figure 6 shows the DMS and DMSPp concentration variabilities under increased $CO_2$ and temperature. These data illustrated that temperature and pH exhibited a significant interactive effect on DMS and DMSPp productions (p = 0.001). The highest DMS and DMSPp concentrations were observed at 18 ℃ under pH 7.9 and pH 7.7, respectively. This result

indicated that temperature may regulate the effect of pH on DMS and DMSPp productions. Therefore, global environmental change manifesting in OA and warming may not result in a decreased DMS as suggested by the effect of elevated $CO_2$ in isolation.





Experimental results can only be hypothesized because information on specific classification of DMS-utilizing bacteria in the experiment is not available. Although such variability in DMS, DMSP, and DMSO concentrations existed during the short-term incubation experiment, whether the differences can be attributed to the effect of pH still remains unclear. Considering the complex interactive effects of environmental factors, an accurate model with appropriate parameterization of

the environmental factors should be developed, and such model can improve our understanding of the earth system's response to predicted global environmental change.

## 5 Conclusion

This study shows the influence of OA on the phytoplankton community structure and concentrations of Chl-*a*, DMS, DMSP, and DMSO in a deck incubation experiment. Concentrations were monitored over the growth and decline of the bloom.

Compared with ambient $CO_2$ levels, the average concentrations of Chl-*a*, DMS, DMSPp, and DMSOd were reduced by approximately 17%, 19%, 17%, and 16% under MC treatment and by 32%, 35%, 21%, and 16% under HC treatment, respectively. By contrast, DMSPd and DMSOp concentrations showed minimal changes under three conditions. Instead of reduction, the ratios of DMSPt/Chl-*a* and DMSOt/Chl-*a* were respectively 22% and 15% higher under HC treatment than under LC treatment. Furthermore, the oxidation-reduction effects of bacteria also played an important role in concentrations

of dimethylated sulfur compounds. In this study, the reason for the difference of dimethylated sulfur compounds concentrations among three treatments could be mainly due to the combined results of changes in phytoplankton production and bacterial metabolism and activity. Additional experiment about the interaction between OA and environmental factors showed that warming can potentially mitigate the drastic effects of increased $CO_2$ levels on DMS and DMSPp productions. In addition, the photodegradation process of DMS was strongly influenced by solar radiation under high $CO_2$ levels.

The response described above illustrated the effect of a relatively rapid increase of pH to the current phytoplankton community. OA might influence the DMS, DMSP, and DMSO concentrations in oceans. Further research is required to facilitate further understanding of the role of specific phytoplankton and bacteria taxa for dimethylated sulfur compounds production under OA and explain the effect of OA on DMS compound cycling in future acidic ocean.

*Competing interests*. The authors declare that they have no conflict of interest.

*Acknowledgements*. The authors thank the captain and the crew of the R/V "*Run Jiang No. 1*" for help and cooperation during the two cruises. This work was financially supported by the National Natural Science Foundation of China (Grant Nos. 41320104008 and 41576073), the National Key Research and Development Program of China (Grant No.

2016YFA0601301), the National Natural Science Foundation for Creative Research Groups (Grant No. 41521064), AoShan Talents Program of Qingdao National Laboratory for Marine Science and Technology (No. 2015 ASTP).



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





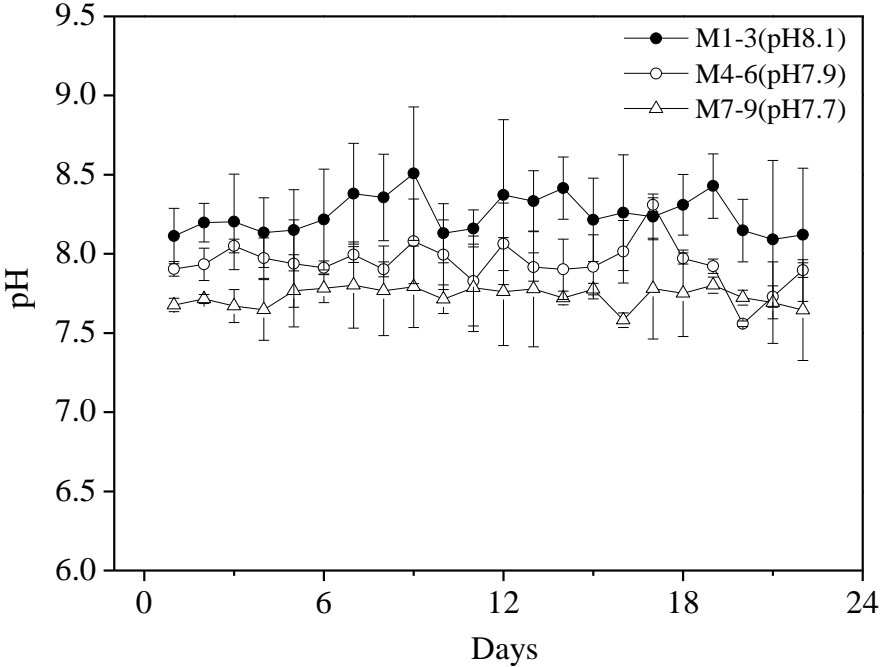

**Figure 1.** Temporal changes in pH for averages of the nine barrels over the triplicates treatments.





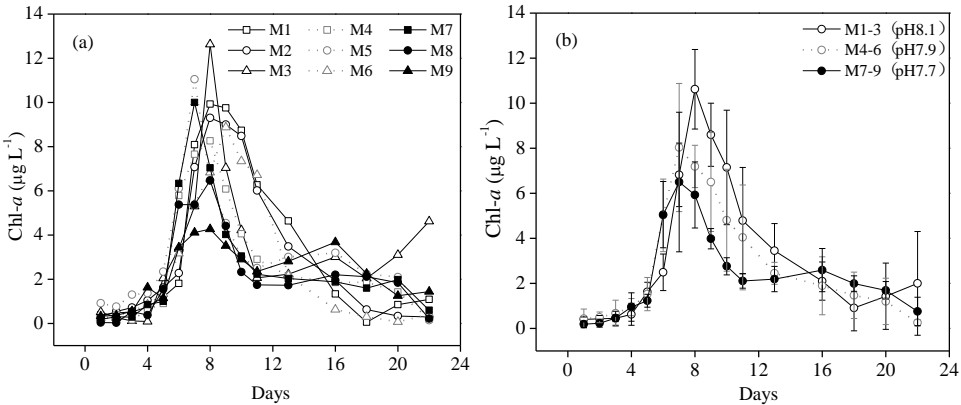

4  **Figure 2.** Temporal changes in the Chl-*a* concentrations (μg L$^{-1}$) for (a) each of the nine barrels and (b) averages over the

5  triplicates treatments.





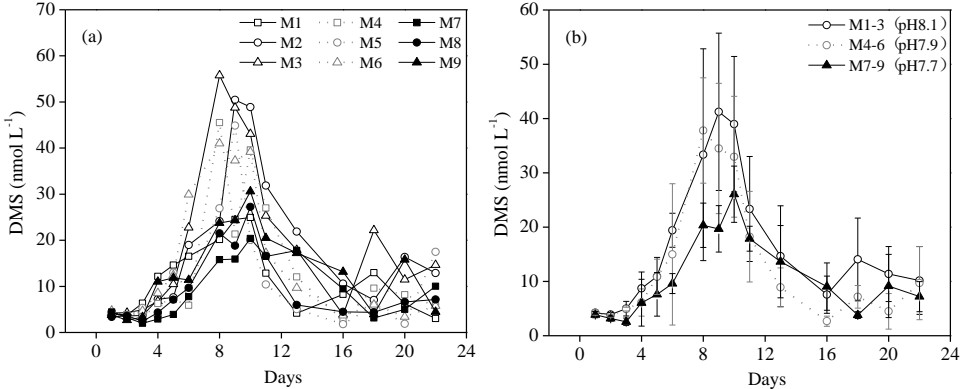

7 **Figure 3.** Temporal changes in the DMS concentrations (nmol L$^{-1}$) for (a) each of the nine barrels and (b) averages over the

8 triplicates treatments.





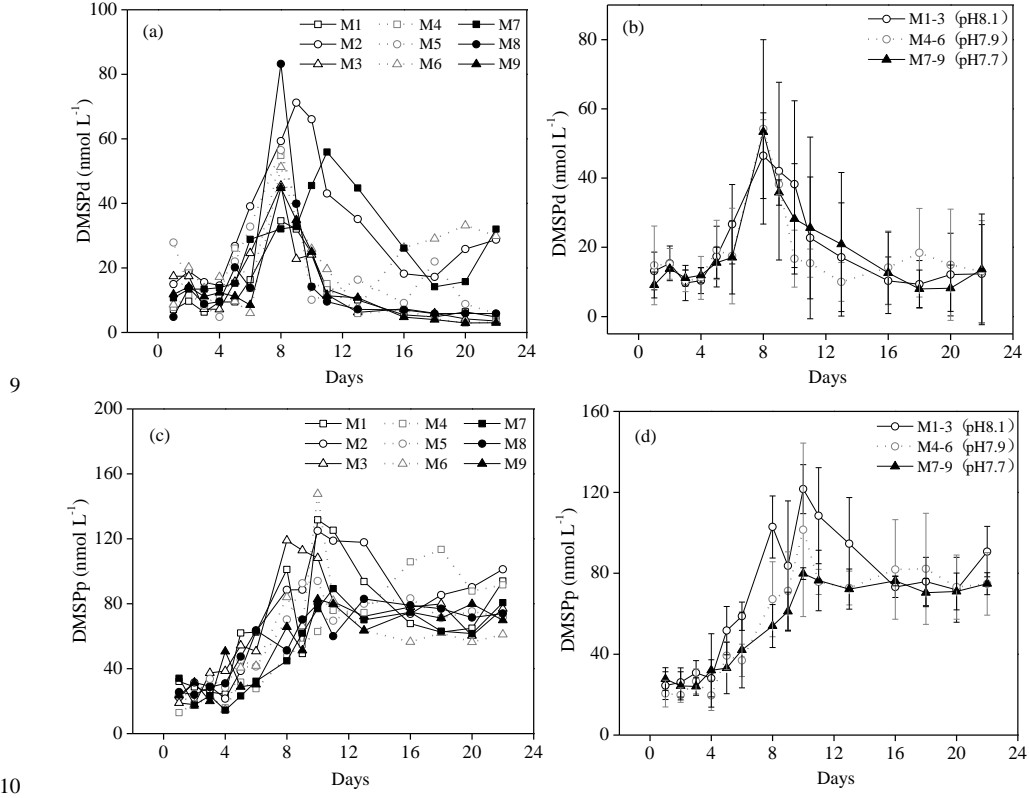

**Figure 4.** Temporal changes in the DMSPd and DMSPp concentrations (nmol L$^{-1}$) for (a, c) each of the nine barrels and (b, d)

averages over the triplicates treatments, respectively.



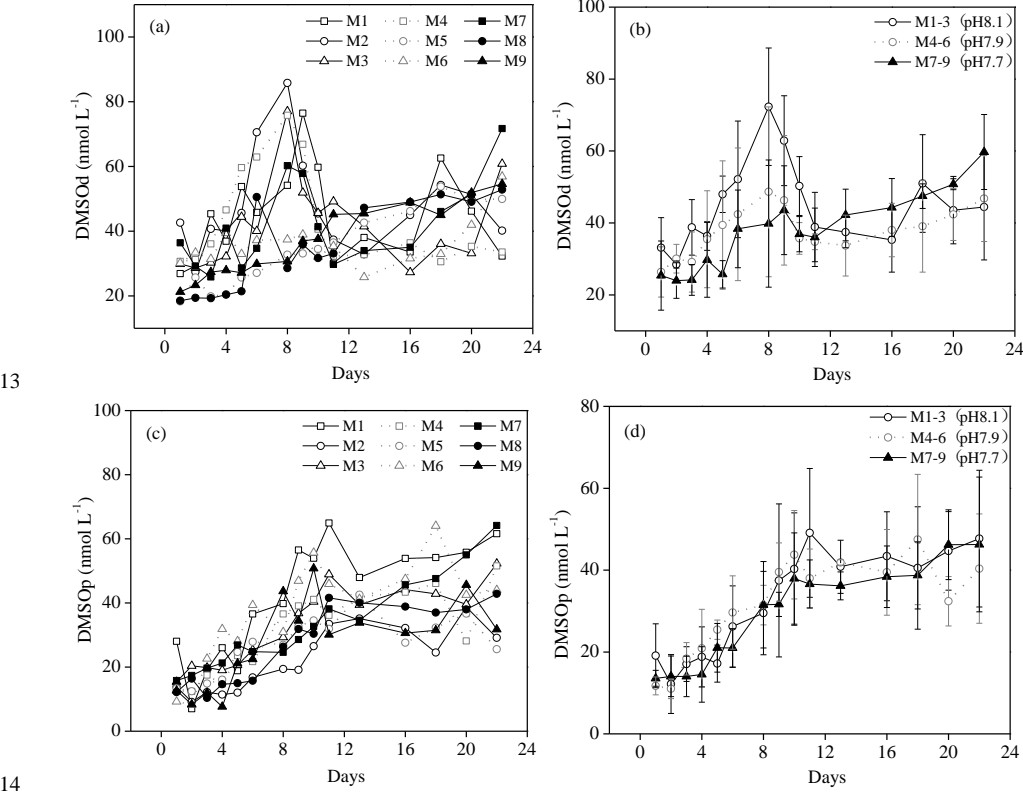

15 **Figure 5.** Temporal changes in the DMSOd and DMSOp concentrations (nmol L$^{-1}$) for (a, c) each of the nine barrels and (b,

16 d) averages over the triplicates treatments, respectively.





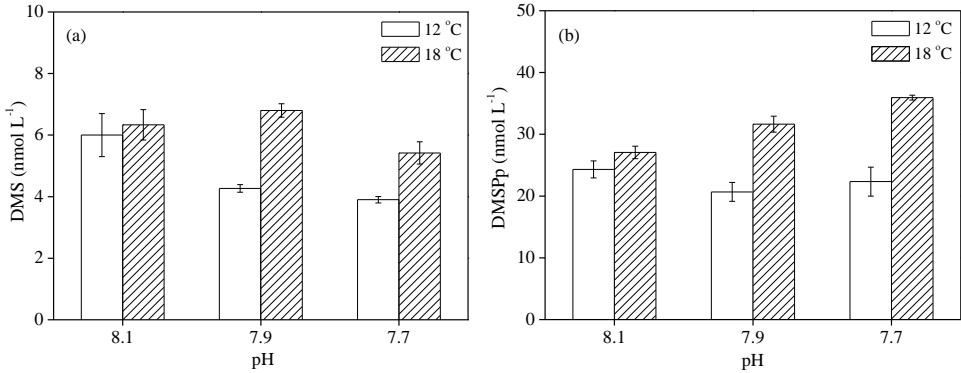

18      **Figure 6.** Average values of (a) DMS and (b) DMSPp concentrations in the cross experiment of OA and temperature, with

19      three replicate incubation samples per treatment.




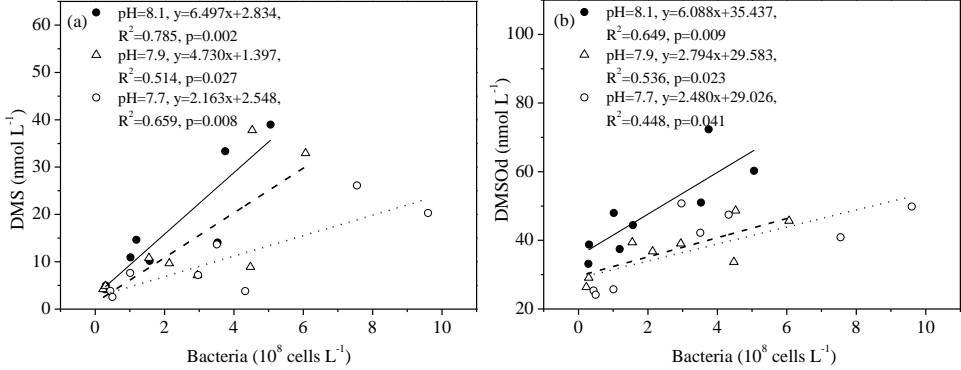

**Figure 7.** Relationships between bacterial abundance, (a) DMS and (b) DMSOd concentrations in the LC (pH 8.1, filled

22    circle, solid lines), MC (pH 7.9, empty triangle, dash lines) and HC (pH 7.7, empty circular, dotted lines) treatments during

23    the experiment.





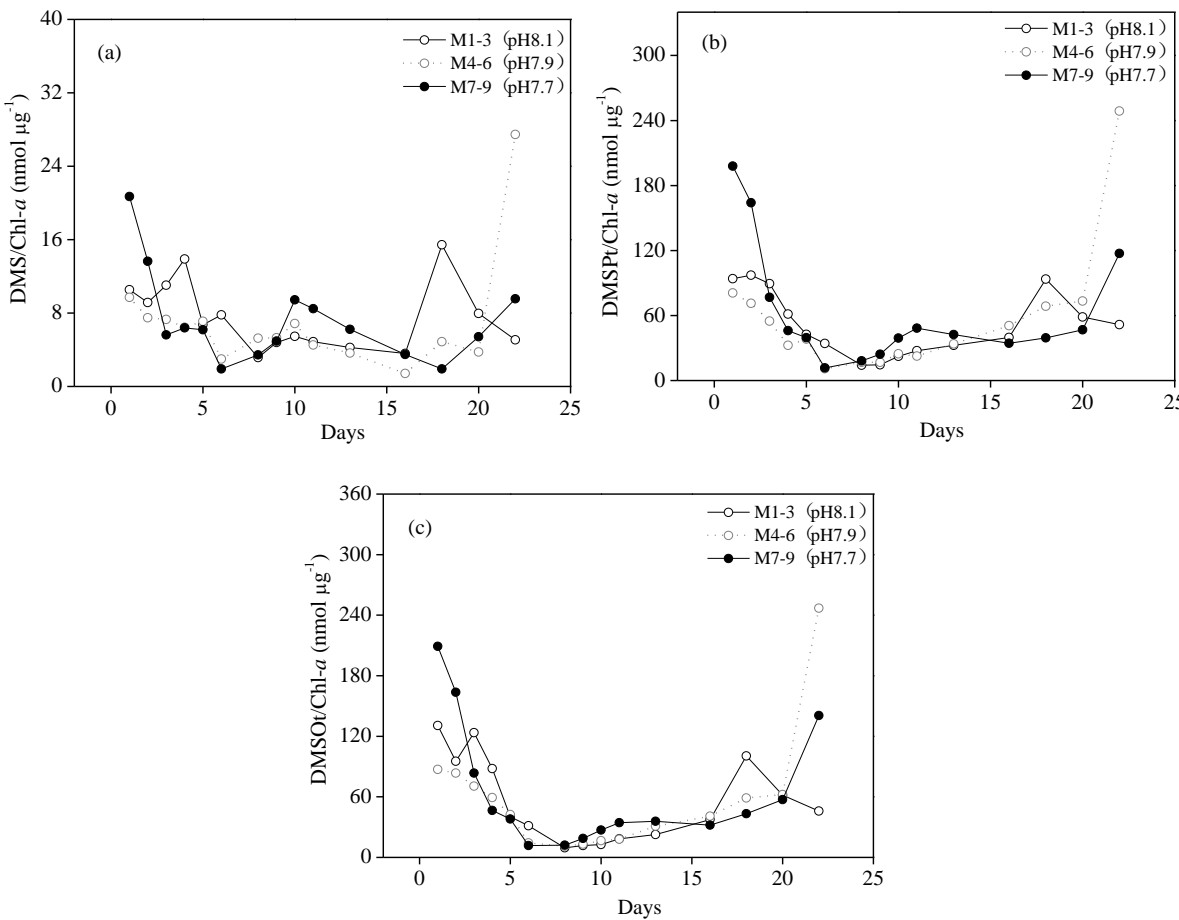

**Figure 8.** Mean ratios of (a) DMS/Chl-*a*, (b) DMSPt/Chl-*a*, (c) DMSOt/Chl-*a* for the LC, MC and HC treatments.



4      **Table 1.** Preliminary carbonate parameters of three treatments during the incubation experiment.

|  | pH | DIC ($\mu$mol kg$^{-1}$) | $p$CO$_2$ ($\mu$atm) | HCO$_3^-$ ($\mu$mol kg$^{-1}$) | CO$_3^{2-}$ ($\mu$mol kg$^{-1}$) |
|---|---|---|---|---|---|
| M1-3 | 8.1 | 2271 | 389.2 | 2087.2 | 168.0 |
| M4-6 | 7.9 | 2263 | 635.9 | 2130.0 | 107.1 |
| M7-9 | 7.7 | 2274 | 1040.7 | 2164.0 | 67.5 |





6 **Table 2.** The mean dimethylated sulfur compounds and Chl-*a* concentrations and bacterial abundance in the different

7 treatments over the entire experiment.

| Treatments (pH ± s.d.) | | Sample concentrations | | |
|---|---|---|---|---|
| | Sample | Minimum | Maximum | Mean ± s.d. |
| M1-3 | Chl-*a* ($\mu$g L$^{-1}$) | 0.06 | 12.6 | 3.14 ±3.22 |
| (pH = 8.11 ±0.017) | Bacterial abundance ($10^8$ cell L$^{-1}$) | 0.29 | 5.06 | 2.09 ±1.78 |
| | DMS (nmol L$^{-1}$) | 3.03 | 55.8 | 16.5 ±12.4 |
| | DMSPd (nmol L$^{-1}$) | 3.54 | 71.2 | 20.2 ±12.5 |
| | DMSPp (nmol L$^{-1}$) | 17.9 | 132 | 69.5 ±31.9 |
| | DMSOd (nmol L$^{-1}$) | 26.9 | 85.8 | 44.9 ±11.7 |
| | DMSOp (nmol L$^{-1}$) | 6.99 | 64.9 | 32.3 ±12.8 |
| M4-6 | Chl-*a* ($\mu$g L$^{-1}$) | 0.08 | 11.1 | 2.60 ±2.31 |
| (pH = 7.90 ±0.022) | Bacterial abundance ($10^8$ cell L$^{-1}$) | 0.23 | 6.07 | 2.78 ±2.12 |
| | DMS (nmol L$^{-1}$) | 1.79 | 45.5 | 13.4 ±12.0 |
| | DMSPd (nmol L$^{-1}$) | 2.88 | 56.5 | 18.9 ±11.8 |
| | DMSPp (nmol L$^{-1}$) | 12.9 | 148 | 57.5 ±27.3 |
| | DMSOd (nmol L$^{-1}$) | 18.3 | 75.8 | 37.9 ±6.64 |
| | DMSOp (nmol L$^{-1}$) | 8.24 | 64.0 | 31.4 ±11.6 |
| M7-9 | Chl-*a* ($\mu$g L$^{-1}$) | 0.03 | 10.0 | 2.14 ±1.72 |
| (pH = 7.72 ±0.029) | Bacterial abundance ($10^8$ cell L$^{-1}$) | 0.44 | 9.60 | 3.74 ±3.35 |
| | DMS (nmol L$^{-1}$) | 1.98 | 30.6 | 10.7 ±7.25 |
| | DMSPd (nmol L$^{-1}$) | 2.93 | 83.2 | 19.0 ±12.4 |
| | DMSPp (nmol L$^{-1}$) | 14.4 | 89.3 | 54.6 ±21.7 |
| | DMSOd (nmol L$^{-1}$) | 18.5 | 71.7 | 37.9 ±10.7 |
| | DMSOp (nmol L$^{-1}$) | 7.67 | 64.2 | 29.4 ±11.9 |



10    **Table 3.** DMS photolysis rate constants and turnover time in the cross experiment of OA and light, with three replicate

11    incubation samples per treatment.

| pH | $K_{natural\ light}$ (d$^{-1}$) | $\tau_{photo}$ (d) | $K_{UVB}$ (d$^{-1}$) | UVB contributions | $K_{UVA}$ (d$^{-1}$) | UVA contributions | $K_{visible\ light}$ (d$^{-1}$) | Visible light contributions |
|---|---|---|---|---|---|---|---|---|
| 8.1 | 4.02 | 1.19 | 1.13 | 28.0% | 2.37 | 59.1% | 0.52 | 12.9% |
| 7.9 | 5.19 | 0.92 | 3.20 | 61.7% | 1.53 | 29.5% | 0.46 | 8.80% |
| 7.7 | 6.32 | 0.76 | 4.73 | 74.9% | 1.35 | 21.3% | 0.24 | 3.83% |
| Average | 5.18 | 0.96 | 3.02 | 55.0% | 1.75 | 37.0% | 0.41 | 9.00% |



14    **Table 4.** Pearson's correlation coefficient and associated significance level for mean dimethylated sulfur compounds and

15    Chl-*a* concentrations under three different $CO_2$ treatments.

|  | Chl-*a* | DMS | DMSPd | DMSPp | DMSOd | DMSOp |
|---|---|---|---|---|---|---|
| **LC (pH = 8.1)** | | | | | | |
| Chl-*a* | 1 | | | | | |
| DMS | **.925**[**] | 1 | | | | |
| DMSPd | **.946**[**] | **.923**[**] | 1 | | | |
| DMSPp | **.724**[**] | **.732**[**] | .570[*] | 1 | | |
| DMSOd | .791[*] | **.778**[**] | **.789**[**] | .522[*] | 1 | |
| DMSOp | .334 | .380 | .119 | **.816**[**] | .200 | 1 |
| **MC (pH = 7.9)** | | | | | | |
| Chl-*a* | 1 | | | | | |
| DMS | **.913**[**] | 1 | | | | |
| DMSPd | **.769**[**] | **.784**[**] | 1 | | | |
| DMSPp | .409 | .445 | .198 | 1 | | |
| DMSOd | .541[*] | .533[*] | .626[*] | .481 | 1 | |
| DMSOp | .400 | .382 | .137 | **.929**[**] | .545[*] | 1 |
| **HC (pH = 7.7)** | | | | | | |
| Chl-*a* | 1 | | | | | |
| DMS | **.657**[**] | 1 | | | | |
| DMSPd | **.756**[**] | **.806**[**] | 1 | | | |
| DMSPp | .323 | .570[*] | .218 | 1 | | |
| DMSOd | .308 | .238 | .080 | **.807**[**] | 1 | |
| DMSOp | .265 | .439 | .157 | **.951**[**] | **.896**[**] | 1 |

16    **. Correlation is significant at the 0.01 level (2-tailed).

17    *. Correlation is significant at the 0.05 level (2-tailed).

