# Peer review of "Effects of the interaction of ocean acidification, solar radiation, and warming on biogenic dimethylated sulfur compounds cycling in the Changjiang River Estuary"

_Biogeosciences, 2017_

## Referee Comment (RC1) · Anonymous Referee #1 · 23 Nov 2017

The Ocean Acidification (OA) community has seen a surge in research in the past 10-15 years, with several hundreds of quality papers being published each year. More recently, the OA community has turned its attention to a very important question: the OA problem in the context of multiple stressors (temperature, light, nutrients, etc) facing oceanic ecosystems. This issue is fundamentally important but raises questions that are not simple to answer; effective experimental designs are not easy to recreate, the oceanic carbonate system remains a challenge to tackle, and the complexity of statistical analysis related to multiple stressor experiments heightens as stressors

are added to the mix. The challenge is even greater when attempting to undertake a study focused on the impacts of declining pH on climate-active gases that come with their own complex dynamics. However, such undertakings are necessary if we are to increase our level of confidence on the biogeochemical responses, interactions and potential retroactions of these climate-relevant biogenic compounds under OA in a rapidly changing climate. The paper "Effets of the interaction of ocean acidification, solar radiation, and warming on biogenic dimethylated sulphur compounds cycling in the Changjiang River Estuary" is full of promise towards this goal. However I have many concerns as to the soundness of the scientific approach, experimental design, and statistical scheme used. On several occasions, the lack of clarity in descriptions, methods used, and lack of information altogether impedes the comprehension of the objectives and conclusions of this study. The numerous short falls concern both the core content of the research as well as the form of the paper.

1. On the core.

1.1. Questionable methodological approach.

The methodological approach used by the authors is incomplete and raises several concerns. The most alarming one is trying to report the impacts of OA on the dynamics of biogenic sulphur compounds and having only this to say about carbonate system measurements and monitoring throughout the 23-day experiment: p.3 Lines 29-30 "Seawater pH was constantly measured using a pH meter (. . .) and the precision is +/- 0.002." Have the authors taken into account a correction factor based on fluctuating temperature? pH meters are usually calibrated at 25oC and substantial variation in measurements can ensue from variability in temperature. The authors mention that ". . .temperature was continuously controlled by circulating in situ seawater, hot water, and ice.", suggesting that potentially significant variation in temperature occurred during 23 days. Have they monitored salinity throughout the experiment? The pH scale (National Bureau of Standard scale (pHNBS), free scale (pHF), total scale (pHT), or seawater scale (pHSWS)) is not mentioned in the manuscript.

To ensure reproducibility, it is critical to report and monitor at least two variables of the carbonate system of seawater (as well as salinity and temperature) for the entire period of the experiment. The authors do add a separate paragraph at the end of the Analytical procedures section where they mention the following: p.5 Lines 9-12 "Total dissolved inorganic carbon (DIC) was determined by a DIC analyzer (AS-C2, Apollo SciTech Inc., Georgia, USA). 10 A sample of 0.5 mL was acidified by 0.5 mL 10% phosphoric acid and then the extracted $CO_2$ gas was measured using a nondispersive infrared (NDIR) $CO_2$ detector (LI-6262, Li-COR Inc., USA) with a precision of 0.1%. The total alkalinity (TA) and $pCO_2$ in sea water were calculated from DIC and pH using the CO2SYS (as refitted by Dickson and Millero (1987))." However these measurements seem to have been made only at the beginning of the experiment for three treatments (as seen in Table 1. Preliminary carbonate parameters of three treatments during the incubation experiment) where no replication is shown and where no actual temporal monitoring is available. What is the use of having a single measurement of DIC at the beginning of the experiment with no follow up throughout the experiment? I also question the methods used to measure DIC, do the authors have a reference to cite here? 0.5ml seems like a very small volume for the measurement of DIC. 125 to 250ml samples are usually taken to accurately measure total inorganic carbon.

Furthermore, the authors do not mention the formulations used to calculate all variables with CO2SYS: concentrations of total boron, $CO_2$ solubility (K0), Dissociation constants of carbonic acid (K1 and K2), boric acid (Kb), water (Kw), phosphoric acid (Kp1, Kp2, Kp3), silicic acid (Ksi), hydrogen fluoride (Kf), and bisulfate (Ks), Solubility products of calcite (Kspc) and aragonite (Kspa). A very useful document prepared in the framework of the data management activity of the Ocean Acidification International Coordination Centre of the International Atomic Energy Agency can be found here (OAICCL: www.iaea.org/ocean-acidification) and shares recommendations proposed in the Guidelines for reporting ocean acidification data in scientific journals.

I also worry about trying to measure concentrations of volatile DMS within noncollapsible barrels from which volumes of water are extracted daily creating more and more headspace over the course of 23 days (figure 3). The authors do not offer any explanation for this, or any suggestions as to how they correct for this problem. We learn that the authors used filtered (what porosity? Axenic? 0.2um, 0.7um?) water to run side-incubations to measure the impact of light on DMS. A first thought here is that any type of filtration process may introduce biases. DMSP-producing communities are notoriously sensitive to filtration, the most common problem is the pressure-induced heightening of the DMSPd pool caused by the rupture of healthy cells. This can artificially enhance pools of DMS through mixing of DMSPd and DMSP-lyase enzymes. This is never mentioned. Also, DMS is controlled by both biology (bacteria, viruses, phyto, zoo) and physics (light, wind). Thus the response of DMS to various light regimes may occur through various pathways (photochemically but also through its impact on primary producers or bacterioplankton). The methodological scheme suggests that the authors here are only investigating the impact of pH and light on photolysis rates of DMS. Yet it is unclear if bacteria/phyto are still present in these experiments (what is the porosity of the filter used?). On page 8 line 25 the authors state the following: "In order to assess the community-level response to the ocean change in future $CO_2$ and light conditions, photolysis rate constants (K, d−1) for DMS were calculated." How is it possible to assess a "community-level" response when filtered water is used presumably to focus on physical processes only (photochemistry)? The authors state objectives that cannot be answered through the methods they use.

The authors use 20L barrels to run their 23 day incubation experiment. Of those 23 days we learn that 18 are onboard the ship and 5 are conducted in the lab (page 3 line 16)? This is very worrisome. No further information is given here. How were the barrels moved? Presumably a lot of mixing occurred during the transit? How were temperatures kept constant during this time? For an experiment looking to identify the impacts of light and temperature, (and pH) on the dynamics of a DMSP-DMS-producing community, this major shift in environmental settings introduces a lot of potential variability that can obscure the response. Yet this is simply glanced over and never mentioned

again. Also, I am curious to know (this could be calculated from the methodologies) how much water was left in the 20L barrels prior to the last sampling day? At least 10L? Or less?

1.2 The authors treat the response of a complex Estuary to OA the same way they would the response of open waters to OA.

The OA problem is increasingly complex in estuarine and coastal waters where freshwater runoff, tidal mixing and high biological activity contribute to variations in $CO_2$ and pH on different time scales. The surface mixed-layer $pCO_2$ can vary spatially and is strongly modulated by biological productivity during the phytoplankton growth season. Surface $pH_T$ in Estuaries can also vary significantly within a single tidal cycle, nearly as much as the world's ocean have experienced in response to anthropogenic $CO_2$ uptake over the last century... Studying the impact of OA in these circumstances (high natural variability in pH and possible resilience of communities) is not devoid of interest, but the authors do not state this, they don't even mention these natural fluctuations and how these could affect the communities, on a basic level, are these communities already tolerant to rapid fluctuations in pH? They treat the impact of OA on this very complex ecosystem as if it was the impact of OA on open oceanic waters. This is troublesome.

1.3 Statistical approach

The description of the statistical strategies used by the authors is confusing and their application to the dataset is questionable. The title of the paper is full of promise "Effects of the interaction of OA, solar radiation and warming on biogenic sulphur compounds cycling..." but fails to deliver on that promise. The methodological and statistical approach proposed by the authors does not allow them to explore potential interactive effects of three stressors on the sources and sinks of DMS and DMSP. Multiple stressors can influence a variable independently (additive), or interact to either reduce (antagonistic) or enhance (synergistic) that variable in a nonlinear, unpredictable fashion. I suggest the authors read Todgham and Stillman (2013), Riebesell and Gattuso (2015), Reum et al. (2015), Gunderson et al (2016). A collapsed factorial approach would have been more informative and simpler to interpret, see Boyd et al (2015). Please also see: "SCOR working group 149 https://scor149-ocean.com Changing Ocean Biological Systems (COBS): How will biota respond to a changing ocean?" It contains pertinent information for moving studies from single to multiple drivers.

On page 12 lines 24-26 the authors mention the following: "The effect of the interaction between OA and environmental conditions complicates the overall ecosystem response. Hence, comprehensive consideration of OA and solar radiation can better interpret and understand feedbacks between OA and global climatic change." A very small exploration of interactive effects is shown in Figure 6 for pH and temperature, and there appears to be a subset of information on pH and certain measurements of light in Table 3, but the interactive impacts of solar radiation, pH and temperature together are not explored. I do not see any "comprehensive consideration". Table 3 seems to relate one type of pH with one type of light treatment, but no combinations... pH 8.1 + light control (quartz), pH 8.1 + Mylar D (UVA), pH8.1 + Plexiglass (PAR), pH8.1 + Dark. pH + light control (quartz), pH 8.1 + Mylar D (UVA), pH8.1 + Plexiglass (PAR), pH8.1 + Dark. Same with pH 7.9 or 7.7 with various light treatments. It is not clear what the rational is behind choosing a specific pH with a specific light treatment (?).

The authors mention the following on page 3 lines 11-13: "Moreover, we examined the photolysis rate and concentration changes of DMS under the dual stressors of changing pH and solar radiation/temperature and assessed the coupling effects of OA, solar radiation and warming on biogenic dimethylated sulfur compounds cycling." Where is the basic information on light? How was light measured? How did it vary naturally over the 23 day experiment? Where is the basic information on temperature? How was it measured? How did it vary over time? I do not see this anywhere. This is very concerning. Furthermore, the authors seem to treat light and temperature as a single

fixed factor (light/temperature) (? I may be wrong because the text is convoluted and unclear) and proceed with a two-way ANOVA to disentangle the impacts of pH, light, and temperature on concentrations of sulphur compounds? Ecologically and statistically, this is difficult to justify. pH + Light experiments were carried out under natural ambient solar radiation (lines 139-140) with 3 treatments and 1 control (full light) and then information is inferred for a 5th variable by subtracting the effects of one treatment from another treatment. The authors state that they also ran pH + Temperature experiments (with 1 treatment (+4oC) and control but it sounds like these are separate experiments from the light + pH experiments, however it is rather unclear): Lines 140-142: "Temperature experiments were also carried out with unfiltered water in quartz bottle under in situ temperature at 12 ïČřC and high temperature at 18 ïČřC for 8 h. The temperature was continuously controlled by circulating in situ seawater, hot water, and ice." (This alone begs the question of the uniformity of the temperature treatment itself if ice and hot water are necessary to stabilize the temperature (there were likely a lot of fluctuations, yet these are not reported)). In any case, I have serious doubts as to the validity of lumping together Light and Temperature variables into a single fixed factor as it seems to be proposed in the statistical paragraph of the methodological section. Also are the underlying assumptions of normal distribution for the response variables (S compounds) respected? There is no mention of this. The authors state the following at lines 25-28 p.5: "Seawater pH was adjusted with CO2-saturated seawater during the experiment to maintain a stable pH environment. In the first week, pH level was comparatively stable with a better control, but pH fluctuated obviously and was difficult to control in the stable and decline phases of algal growth. As shown in Fig. 1, the pH showed relatively apparent fluctuations on days 9-12 and days 18-20." Again, it is difficult here to conceive that pH was a fixed explanatory variable. Judging by Figure 1, there are several instances when there doesn't appear to be a statistical difference between the pH treatments with very large error bars.

On page 5 Lines 14-20, the authors describe very briefly the statistical approaches used, but these spark more questions than answer anything. "Statistical analysis was

performed using SPSS version 18.0 (SPSS Inc., IBM, USA). Pearson's correlation coefficient and probability (p) were calculated to evaluate the quality of the fit when variables were normally distributed. (How was normality assessed?) T-test was used to determine whether a significant difference existed between two treatments. (WHICH treatment? More information is needed.) "Variability in the concentrations of biogenic dimethylated sulfur compounds was analyzed using two-way ANOVA with pH and temperature/light as fixed factors and concentration as a random factor to understand whether an interaction existed between pH and temperature/light on concentration. (Are the authors suggesting that temperature/light are one fixed factor?) Value at $p \leq 0.05$ was considered statistically significant. Linear correlation analyses were used to determine the response of DMS, DMSO concentrations, and bacterial abundance to OA using Origin 9.1." The authors proceed to establish Pearson's correlations between two variables within a same pH condition. I am not convinced that this is useful. What is the purpose of this exactly? How does it inform us of the complex response of these variables to fluctuations of pH itself? Overall, the statistical schemes proposed by the authors do not seem appropriate, and they do not facilitate the interpretation of possibly complex responses. Table 4 presents these correlation coefficients. What are the degrees of freedom (DF) for these analyses? Why is a coefficient of 0.791 (between DMSOd and chla) significant at the p <0.05 level while a coefficient of 0.778 (between DMSOd and DMS) is significant at the p <0.01 level ? More information is needed about the n and the DF. On pages 10 and 11, the authors discuss these correlations between sulphur compounds and other variables (within a same pH treatment) at length but I am not convinced that these are very informative on the impact of pH itself on the sulphur compounds. When looking at figure 2-3-4-5, the error bars are so wide for the pH treatments (substantial overlapping most of the time, ex: DMSOp), that it is hard to understand what the purpose of these inner-treatment correlations is. A very convoluted text does not help in the matter. An example of this here: "A significant correlation between DMS and DMSOd was observed in LC treatments (Table 4). The result was in accordance with the findings of Hatton et al. (2004) and Zindler-Schlundt

et al. (2015) who showed that DMSO and DMS are closely related because of the direct formation of DMSO via bacterial DMS oxidation. The close link between DMS and DMSOd was not observed in MC and HC treatments. The changes in the production, consumption, and degradation processes which caused by the decreasing pH might mask the relationship between DMS and DMSOd. (Production/consumption and degradation of what exactly?) In addition, DMSPd concentration showed an obvious correlation with DMSOd only in LC treatments for the entire duration of the experiment (Table 4). DMSPd may be cleaved to DMS by specific bacteria that contain DMSP lyase (Curson et al., 2008), followed by the bacterial oxidation of DMS to DMSOd. This relationship was not found in MC and HC treatments. The phenomenon might be caused by the change in phytoplankton community and bacterial oxidation of DMS to DMSO." It is very difficult to follow the logic here. What change in phyto community? What are the authors talking about? They do not show any information on community composition for this study. And this is another important shortcoming of this paper: no phytoplankton identification. DMSP/DMS cycling is intimately linked with species composition, yet we have no idea WHO is there and HOW pH may affect the primary producers responsible for a substantial part of the DMS/DMSP cycling. In the introduction the authors mention this: "Shaw hypothesized that DMS and sulfate aerosols are linked to global climate. This link was further elaborated by Charlson et al. (1987). Consequently, ocean acidification (OA)-induced changes in the primary productivity might impact on the production rate and sea-to-air emission of DMS and these impacts might further affect cloud formation and climate". The authors do not explore the primary production side of things at all. There are no PP rates, no phyto identification, only chla which is a proxy of biomass that does not inform us on whether pH impacts chla levels through physiological processes or through variability in the species composition.

1.4 Out-dated information and statements that are too general.

Page 1 line 28. The information given here is not up to date. The authors reference a paper written in 2000. Progress has been made in the last ca. 15 years. Anthropogenic

carbon dioxide (CO2) emissions have increased atmospheric CO2 concentrations from their pre-industrial value of 280 to ca. 400 $\mu$atm in 2016 (NOAA-ESRL).

Page 1 lines 29-30. Under which scenario exactly? Please be precise and use the latest information available. Concentrations of 850 to 1370 $\mu$atm are expected by the end of the century under the business-as-usual scenario RCP 8.5 (IPCC, 2013).

Page 1 line 30 and beyond. The authors don't offer any explanation, even if short, as to the underlying processes involved in pH modulation in oceans as a result of CO2 increases in the atmosphere. They should spend a little more time explaining this. That is one the focuses of their paper after all. Also there is a more recent paper by Caldeira and Wickett (2005) suggesting that the surface ocean pH is expected to decrease by an additional 0.3-0.4 units under the RCP 8.5 scenario by 2100.

Page 2 lines 1-2. What is the reference here? There are several consequences of increasing OA. Modifications in DIC is the primary, from which modifications in carbonate system ensue (which includes modifications in saturation states of calcite and aragonite, which by the way the authors don't explain the importance of: why talk about calcite and aragonite?). Modifications in the amount of protons (H+) is another. Etc. Etc.

Page 2 lines 2-4. This is quite a general statement. What changes exactly (less calcite, more H+, more CO2?) will affect what physiological processes exactly, and what type of marine "organism" exactly? There is so much literature on the many aspects of OA and the consequences for marine organisms (Calcifiers? Phyto? Zoo? Fish? Etc), and the potential impacts may be positive, negative, resilience etc... The phrase written is simply too generalist, it should offer at least a glimpse into the multi-tiered discoveries made by the OA research community, or at least focus on the aspects that are relevant to the author's research: DMS-producing microbial communities...

Page 2 lines 19-20. The authors make very general statements without offering much information: "The conversion of DMSP to DMS is controlled by a number of chemical

and biological processes." Yes. But which ones?

These are only some examples of the generalities, more can be found in the manuscript.

2. On the form

As it currently stands, the overall level of language does not meet the high standards of BG. There are countless examples of awkward formulations and missing words throughout the paper, including: "...more CO2 (low pH) will affect physiological process of marine organism." Page 2 line 3 Page 2 lines 5 and beyond : The transition between the impact of OA on they physiology of marine organisms and the oceanic production of climate-relevant gases is not well established.

"Once emitted to atmosphere, DMS. . .". Page 2 line 10 "DMSO was initially conceived as a sink for DMS. Nevertheless, DMSO was later found a potential source of DMS." Page 2 lines 26-27 "The ocean undergoes multiple environmental changes. Other climatic ecological stressor or factors would probably alter the effects of ocean acidification on the production and consumption process of DMS in both direct and indirect ways." Page 3 lines 2-4 "The photochemical process of DMS in the surface water would change due to the changing light level and seawater pH level." Page 3 line 5

There are so many more examples. I will stop here because I believe a profound language editing needs to be conducted and this goes beyond the scientific mandate of a reviewer.

The introduction offers many general statements that only skim a fraction of the inherent complexity and wealth of information related to OA research. The paper lacks coherence and clarity and there is redundancy in the writing. Formulation of phrases is awkward, making the text very difficult to follow.

Page 2 lines 6-8: Redundancy in the phrase: "It is produced by enzymatic cleavage of dimethylsulfoniopropionate (DMSP) (Gabric et al., 2010), which is synthesized by

marine phytoplankton as a phytoplankton-derived precursor of DMS."

Page 2 lines 16-17: Redundancy in the phrase as well as unclear: "DMSP, as the main precursor of DMS, . . ., presents an important effect on the biogeochemical cycle of climatically active trace gas DMS." This is rather vague and redundant. Needs rephrasing.

Page 2 line 21 The word "preferred" is not appropriate.

Page 2 line 26: "DMSO was originally conceived as a sink for DMS." This phrase does not make sense. Needs rewording.

Again, these are only examples, more can be found in the manuscript.

In short, at this point, there is much work to be done before this paper can be considered for publication in BG. First and foremost, there are several uncertainties and questions related to the methodologies used and the experimental design itself. Without a sound experimental plan and measurements, the rest of it (interpretations and conclusions) is useless. Overall, the entire methodological aspects related to the OA part of the experiments are unclear and lacking (carbonate system measurements absent, salinity?, temperature records?). It is unclear whether the temperature and light treatments were separate experiments. And where are those measurements of temperature and light? What about primary production, phytoplankton species composition?The statistical schemes do not seem appropriate. The objectives of the study are not well stated (I am referring here to the Estuarine context in which the experiment takes place). The environmental setting is not clearly established; the authors don't even situate the study area on a map. The level of language is not up to par.

---

## Referee Comment (RC2) · Anonymous Referee #2 · 14 Dec 2017

I must start by recognizing the efforts of Anonymous Reviewer #1 – a very detailed, fair and constructive review which provides the authors with a comprehensive understanding of the status of this paper.

Importantly, I would like to stress that I unequivocally agree with Reviewer #1 on all points. Unfortunately, this paper fails to deliver on the promise of its title. In its current form, it falls far short of the standards of Biogeosciences and would require some significant and extensive re-working to be considered suitable. I don't feel that another detailed review would useful for the authors, so I will briefly cover a few points without

repeating the suggestions and comments of Reviewer #1.

1. Lack of reference to the existing literature: There are now a good number of published papers out there that report the effects of OA on DMS and related compounds, but the authors have only cited a handful. In particular, the works of Archer et al. (Arctic mesocosm), Webb et al. (various mesocosms), and Hopkins et al. (various mesocosms and shipboard studies), Hussherr et al. (shipboard) are not mentioned at all, which seems a little odd. To me, it is important to place your findings within the context of the existing literature. They do mention a few examples, but with some errors in referencing: e.g. Avgoustidi et al. 2012 is given in the reference list, but they also mention Avgoustidi 2006 in the text but not the list of references.

2. Lack of appreciation of the 'bigger picture': Why do we perform these kinds of experiments? Ultimately, it is to generate data that may be utilised by modellers within earth system models. Some recognition/discussion of this would be useful. At least, some discussion of published studies (Six et al., Schwinger et al.) to provide the reader with an understanding of the potential DMS-climate feedbacks resulting from OA. Otherwise why do we care?

3. Language weaknesses: The entire paper needs language checking. The structure of the paper needs some consideration – for example, large chunks of the 'Results' text would probably be considered more suitable for the 'Discussion'.

I would recommend that the authors carefully consider all of the above and the detailed points raised by Reviewer #1 before even considering resubmitting this paper. In its current form, it is far from suitable for publication.

---

## Author Comment (AC1) · 3 Feb 2018

Question 1. On the core. 1.1 Questionable methodological approach The methodological approach used by the authors is incomplete and raises several concerns. The most alarming one is trying to report the impacts of OA on the dynamics of biogenic sulphur compounds and having only this to say about carbonate system measurements and monitoring throughout the 23-day experiment: p.3 Lines 29-30 "Seawater pH was constantly measured using a pH meter (. . .) and the precision is +/- 0.002." Have the authors taken into account a correction factor based on fluctuating temperature? pH

meters are usually calibrated at 25oC and substantial variation in measurements can ensue from variability in temperature. The authors mention that ". . .temperature was continuously controlled by circulating in situ seawater, hot water, and ice.", suggesting that potentially significant variation in temperature occurred during 23 days. Have they monitored salinity throughout the experiment? The pH scale (National Bureau of Standard scale (pHNBS), free scale (pHF), total scale (pHT), or seawater scale (pHSWS)) is not mentioned in the manuscript.

Reply: Thanks for the reviewer's comment. Yes, we have taken into account a correction factor based on fluctuating temperature. pH meters were calibrated at 25 oC in the determination and three parallel samples were measured for each pH value. Temperature was continuously controlled by circulating in situ seawater, hot water, and ice in the temperature experiments for 8 h (not the incubation experiment of ocean acidification for 23 days). In the incubation experiment of ocean acidification (23 days), all incubation barrels were exposed to sunlight in a flow-through water bath to maintain treatments at their in situ temperature. Unfortunately, the salinity data were not determined throughout the experiment. According to the reviewer's suggestion, we have added the relevant content in the revised manuscript. "Seawater pH was continually measured using a pH meter (S210 SevenCompact™, Mettler, Germany, precision: ± 0.002), using the National Bureau of Standards (NBS) scale to calibrate the pH electrode." (Page 4 Line 9-11)

Question: To ensure reproducibility, it is critical to report and monitor at least two variables of the carbonate system of seawater (as well as salinity and temperature) for the entire period of the experiment. The authors do add a separate paragraph at the end of the Analytical procedures section where they mention the following: p.5 Lines 9-12 "Total dissolved inorganic carbon (DIC) was determined by a DIC analyzer (AS-C2, Apollo SciTech Inc., Georgia, USA). 10 A sample of 0.5 mL was acidified by 0.5 mL 10% phosphoric acid and then the extracted $CO_2$ gas was measured using a nondispersive infrared (NDIR) $CO_2$ detector (LI-6262, Li-COR Inc., USA) with a precision of

0.1%. The total alkalinity (TA) and pCO2 in sea water were calculated from DIC and pH using the CO2SYS (as refitted by Dickson and Millero (1987)).” However these measurements seem to have been made only at the beginning of the experiment for three treatments (as seen in Table 1. Preliminary carbonate parameters of three treatments during the incubation experiment) where no replication is shown and where no actual temporal monitoring is available. What is the use of having a single measurement of DIC at the beginning of the experiment with no follow up throughout the experiment? I also question the methods used to measure DIC, do the authors have a reference to cite here? 0.5ml seems like a very small volume for the measurement of DIC. 125 to 250ml samples are usually taken to accurately measure total inorganic carbon.

Reply: Thanks for the reviewer's comment. Unfortunately, no temporal monitoring is available in our experiment. DIC was measured only at the beginning of the incubation experiment and the Standard Deviations have been added in Table 1 in (Page 31 Line 1). A volume of 0.5 mL is sufficient for the measurement of DIC. We have added references in the revised manuscript. “Cai, W. J., Dai, M., Wang, Y., Zhai, W., Huang, T., Chen, S., Zhang, F., Chen, Z., Wang, Z.: The biogeochemistry of inorganic carbon and nutrients in the Pearl River estuary and the adjacent Northern South China Sea. Contin. Shelf Res., 24, 1301–1319, https://doi.org/10.1016/j.csr.2004.04.005, 2004” (Page 15 Line 25-27) “Cai, W. J., Wang, Y.: The chemistry, fluxes, and sources of carbon dioxide in the estuarine waters of the Satilla and Altamaha Rivers, Georgia, Limnol. Oceanogr., 43, 657–668, https://doi.org/10.4319/lo.1998.43.4.0657, 1998.” (Page 15 Line 28-29)

Question: Furthermore, the authors do not mention the formulations used to calculate all variables with CO2SYS: concentrations of total boron, CO2 solubility (K0), Dissociation constants of carbonic acid (K1 and K2), boric acid (Kb), water (Kw), phosphoric acid (Kp1, Kp2, Kp3), silicic acid (Ksi), hydrogen fluoride (Kf), and bisulfate (Ks), Solubility products of calcite (Kspc) and aragonite (Kspa). A very useful document prepared in the framework of the data management activity of the Ocean Acidification International Coordination Centre of the International Atomic Energy Agency can be found here (OAICCL: www.iaea.org/ocean-acidification) and shares recommendations proposed in the Guidelines for reporting ocean acidification data in scientific journals. I also worry about trying to measure concentrations of volatile DMS within non collapsible barrels from which volumes of water are extracted daily creating more and more headspace over the course of 23 days (figure 3). The authors do not offer any explanation for this, or any suggestions as to how they correct for this problem. We learn that the authors used filtered (what porosity? Axenic? 0.2um, 0.7um?) water to run side-incubations to measure the impact of light on DMS. A first thought here is that any type of filtration process may introduce biases. DMSP-producing communities are notoriously sensitive to filtration, the most common problem is the pressureinduced heightening of the DMSPd pool caused by the rupture of healthy cells. This can artificially enhance pools of DMS through mixing of DMSPd and DMSP-lyase enzymes. This is never mentioned. Also, DMS is controlled by both biology (bacteria, viruses, phyto, zoo) and physics (light, wind). Thus the response of DMS to various light regimes may occur through various pathways (photochemically but also through its impact on primary producers or bacterioplankton). The methodological scheme suggests that the authors here are only investigating the impact of pH and light on photolysis rates of DMS. Yet it is unclear if bacteria/phyto are still present in these experiments (what is the porosity of the filter used?). On page 8 line 25 the authors state the following: “In order to assess the community-level response to the ocean change in future CO2 and light conditions, photolysis rate constants (K, d−1) for DMS were calculated.” How is it possible to assess a “community-level” response when filtered water is used presumably to focus on physical processes only (photochemistry)? The authors state objectives that cannot be answered through the methods they use.

Reply: Thanks for the reviewer's comment. We agree with the reviewer's comment. CO2SYS: The value of K0 (the solubility coefficient of CO2) and the conversion between the fugacity and the partial pressure of CO2 are from Weiss, R. F., Marine Chemistry 2:203-215, 1974. The value of Kb (for boric acid), in constant choices 1 to 5, is

from Dickson, Andrew G., Deep-Sea Research 37:755-766, 1990. The boron concentration in constant choices 1 to 5 is from Uppstrom, Leif, Deep-Sea Research 21:161-162, 1974. Values of KW (for H2O), KP1, KP2, and KP3 (for phosphoric acid), and KSi (for silicic acid) are from (in constant choices 1 to 5) Millero, Frank J., Geochemica et Cosmochemica Acta 59:661-677, 1995. K1 and K2 of carbonic acid are from the data that refit by Dickson and Millero (1987) on the seawater scale. In the present study, we measured the DMS concentration in solution after the gas-liquid equilibrium. Changes in DMS production by phytoplankton will result in a simultaneous change in DMS concentration in the solution after the equilibration between solution and headspace, so the presence of the headspace will not affect the overall trend of DMS concentration changes. The seawater using in the photodegradation experiment was filtered with 0.4 and 0.2 $\mu$m Whatman GF/F filter and all equipment were sterilized during the experiment. In addition, the DMS samples were collected and measured without any filtration process. We have added the relevant content in the revised manuscript. "In the separate photodegradation experiment, 1 L seawater sample was filtered directly into precleaned Qorpak bottles through 0.45 and 0.2 $\mu$m polyethersulfone membranes to remove bacteria, phytoplankton, zooplankton and original DMS. Then the standard solution of DMS was added to the filtered seawater." (Page 4 Line 12-14) Furthermore, we have rephrased the sentence "In order to assess the community-level response to the ocean change in future CO2 and light conditions, photolysis rate constants (K, d−1) for DMS were calculated" in the revised manuscript. "In order to assess the photodegradation process of DMS response to the ocean change in future CO2 and light conditions, photolysis rate constants (K, d−1) for DMS were calculated." (Page 9 Line 18-19)

Question: The authors use 20L barrels to run their 23 day incubation experiment. Of those 23 days we learn that 18 are onboard the ship and 5 are conducted in the lab (page 3 line 16)? This is very worrisome. No further information is given here. How were the barrels moved? Presumably a lot of mixing occurred during the transit? How were temperatures kept constant during this time? For an experiment looking to

identify the impacts of light and temperature, (and pH) on the dynamics of a DMSP-DMS-producing community, this major shift in environmental settings introduces a lot of potential variability that can obscure the response. Yet this is simply glanced over and never mentioned again. Also, I am curious to know (this could be calculated from the methodologies) how much water was left in the 20L barrels prior to the last sampling day? At least 10L? Or less?

Reply: The barrels were moved by a truck. The barrels were covered with lids throughout the experiment, including the transportation process, which can avoid the gas loss caused by mixing. The relevant content has been added in the manuscript. "All barrels were sealed with lids throughout the experiment, including the transportation process". (Page 4 Line 6-7) The transport took about twelve hours, during which time the temperature was controlled by the seawater in the incubation tank to keep the incubation barrels at the same temperature. By calculating, approximately 11.6 L seawater was left in the 20 L barrels on the last sampling day.

Question: 1.2 The authors treat the response of a complex Estuary to OA the same way they would the response of open waters to OA. The OA problem is increasingly complex in estuarine and coastal waters where freshwater runoff, tidal mixing and high biological activity contribute to variations in CO2 and pH on different time scales. The surface mixed-layer pCO2 can vary spatially and is strongly modulated by biological productivity during the phytoplankton growth season. Surface pHT in Estuaries can also vary significantly within a single tidal cycle, nearly as much as the world's ocean have experienced in response to anthropogenic CO2 uptake over the last century. . . Studying the impact of OA in these circumstances (high natural variability in pH and possible resilience of communities) is not devoid of interest, but the authors do not state this, they don't even mention these natural fluctuations and how these could affect the communities, on a basic level, are these communities already tolerant to rapid fluctuations in pH? They treat the impact of OA on this very complex ecosystem as if it was the impact of OA on open oceanic waters. This is troublesome.

Reply: Thanks for the reviewer's comment. We agree with the reviewer's comment and have added relevant content in the revised manuscript. "In recent years, the effects of simulated future CO2-induced seawater acidification on DMS production are still controversial (Avgoustidi et al., 2012; Kerrison et al., 2012; Kim et al., 2010; Vogt et al., 2008) and very few data are available on the OA effects on DMS production in estuaries. The patterns of low pH states observed in complex estuary are largely due to the result of freshwater runoff, natural mixing and high biological activity. By the end of this century, OA may become the dominant process reducing the pH in estuary area (Feely et al., 2010). Studying the impact of OA on the DMS cycle in estuary is very necessary and meaningful (Reum et al., 2015)." (Page 3 Line 15-20). "Feely, R. A., Alin, S. R., Newton, J., Sabine, C. L., Warner, M., Devol, A., Krembs, C., Maloy, C.: The combined effects of ocean acidification, mixing, and respiration on pH and carbonate saturation in an urbanized estuary. Estuar. Coast. Shelf Sci., 88, 442–449, https://doi.org/10.1016/j.ecss.2010.05.004, 2010." (Page 16 Line 21-23) In the acidification incubation experiment, chrysophyceae and cryptophyceae disappeared during HC treatment at the late stage of the experiment, indicating a great influence of OA on their growth. But they made up a smaller percentage of this community and diatoms and dinoflagellates dominated the phytoplankton community. According the result of section 4.1.3, the ratios of DMS/Chl-a, DMSPt/Chl-a, and DMSOt/Chl-a showed no difference in the dimethylated sulfur compounds production in terms of phytoplankton biomass. This result indicated that the relative strength of DMS, DMSP, and DMSO production by the phytoplankton communities has not changed and these communities were tolerant to rapid fluctuations in pH. In order to better understand, we added relevant content in the revised manuscript. "This result indicated that the productions of dimethylated sulfur compounds per unit biomass were not affected by pH perturbations and these communities were tolerant to rapid fluctuations in pH." (Page 12 Line 29-31)

Question: 1.3 Statistical approach The description of the statistical strategies used by the authors is confusing and their application to the dataset is questionable. The title of the paper is full of promise "Effects of the interaction of OA, solar radiation and warming

on biogenic sulphur compounds cycling. . ." but fails to deliver on that promise. The methodological and statistical approach proposed by the authors does not allow them to explore potential interactive effects of three stressors on the sources and sinks of DMS and DMSP. Multiple stressors can influence a variable independently (additive), or interact to either reduce (antagonistic) or enhance (synergistic) that variable in a non-linear, unpredictable fashion. I suggest the authors read Todgham and Stillman (2013), Riebesell and Gattuso (2015), Reum et al. (2015), Gunderson et al (2016). A collapsed factorial approach would have been more informative and simpler to interpret, see Boyd et al (2015). Please also see: "SCOR working group 149 https://scor149-ocean.com Changing Ocean Biological Systems (COBS): How will biota respond to a changing ocean?" It contains pertinent information for moving studies from single to multiple drivers.

Reply: In this study, we explored the potential interactive effects of ocean acidification and solar radiation or warming, the title of the paper may be improperly expressed. Thanks for the reviewer's comment and we have changed the title and related narration in the revised manuscript. "Effects of the interaction of ocean acidification and solar radiation/warming on biogenic dimethylated sulfur compounds cycling in the Changjiang River Estuary" (Page 1 Line 1-3) "Moreover, we examined the photolysis rate and concentration changes of DMS under the dual stressors of changing pH and solar radiation/temperature and assessed the coupling effects of OA and solar radiation/warming on biogenic dimethylated sulfur compounds cycling." (Page 3 Line 22-23) "3.2 DMS and DMSP concentration response to the interaction among OA and solar radiation/warming" (Page 9 Line 15) Thanks to the references recommended by the reviewer. We have added these references and relevant interpretation in the revised manuscript. "So these concomitant global change variables should be taken into consideration to better explain the effect of acidification on DMS (Todgham and Stillman, 2013; Boyd et al., 2015; Riebesell and Gattuso, 2015; Gunderson et al., 2016)." (Page 3 Line 14) "Studying the impact of OA on the DMS cycle in estuary is very necessary and meaningful (Reum et al., 2015)" (Page 3 Line 19-20) "Todgham, A. E., Stillman, J.

H.: Physiological responses to shifts in multiple environmental stressors: relevance in a changing world, Integr. Comp. Biol., 53, 539–544, https://doi.org/10.1093/icb/ict086, 2013." (Page 20 Line 9-10) "Boyd P W, Lennartz S T, Glover D M, Doney, S. C.: Biological ramifications of climate-change-mediated oceanic multi-stressors, Nat. Clim. Change, 5, 71–79, https://doi.org/10.1038/nclimate2441, 2015." (Page 15 Line 18-19) "Riebesell, U., Gattuso, J. P.: Lessons learned from ocean acidification research, Nat. Clim. Change, 5, 12–14, https://doi.org/10.1038/nclimate2456, 2015." (Page 19 Line 13-14) "Gunderson, A. R., Armstrong, E. J., Stillman, J. H.: Multiple stressors in a changing world: the need for an improved perspective on physiological responses to the dynamic marine environment, Ann. Rev. Mar. Sci., 8, 357–378, https://doi.org/10.1146/annurev-marine-122414-033953, 2016." (Page 17 Line 6-8) "Reum J C P, Alin S R, Harvey C J, Bednaršek, N., Evans, W., Feely, R. A., Hales, B., Lucey, N., Mathis, J. T., McElhany, P., Newton, J. Sabine, C. L.: Interpretation and design of ocean acidification experiments in upwelling systems in the context of carbonate chemistry co-variation with temperature and oxygen. ICES J Mar. Sci., 73, 582–595, https://doi.org/10.1093/icesjms/fsu231, 2015." (Page 19 Line 9-12) "In addition, Todgham and Stillman (2013) reported that multiple stressors can influence performance independently (additive) or interact interactively (antagonistic or synergistic). We can infer that OA and warming play a synergistic role in the production of dimethylated sulfur compounds. This result is consistent with that reported by Six et al. (2013) who showed that global warming can be amplified through pH dependence of DMS production." (Page 13 Line 21-25)

Question: On page 12 lines 24-26 the authors mention the following: "The effect of the interaction between OA and environmental conditions complicates the overall ecosystem response. Hence, comprehensive consideration of OA and solar radiation can better interpret and understand feedbacks between OA and global climatic change." A very small exploration of interactive effects is shown in Figure 6 for pH and temperature, and there appears to be a subset of information on pH and certain measurements of light in Table 3, but the interactive impacts of solar radiation, pH and temperature

together are not explored. I do not see any "comprehensive consideration". Table 3 seems to relate one type of pH with one type of light treatment, but no combinations. . . pH 8.1 + light control (quartz), pH 8.1 + Mylar D (UVA), pH8.1 + Plexiglass (PAR), pH8.1 + Dark. pH + light control (quartz), pH 8.1 + Mylar D (UVA), pH8.1 + Plexiglass (PAR), pH8.1 + Dark. Same with pH 7.9 or 7.7 with various light treatments. It is not clear what the rational is behind choosing a specific pH with a specific light treatment (?).

Reply: According to the reviewer's suggestion, we have reworded this sentence in the revised manuscript. "Hence, consideration of the interaction between OA and solar radiation can better interpret and understand feedbacks between OA and global climatic change." (Page 13 Line 14-15) Deal et al. (2005) reported that the contributions of UVA, UVB and PAR to DMS photodegradation were quite different and Bouillon and Miller (2005) mentioned that pH level can influence the photodegradation of DMS. So the contribution of UVA, UVB and PAR to DMS photodegradation under different pH levels may different. Furthermore, the ocean undergoes ocean acidification. To better understand the DMS loss through photodegradation in the future ocean, we designed this experiment with five light conditions (natural light, UVA, visible light, dark, UVB: results for full-spectrum natural light minus those for UVB-filtered light) and three pH levels (ambient level: 8.1; level expected at the end of this century: 7.9; level predicted for the middle of the next century: 7.7). "The research of Deal et al. (2005) mentioned that the photodegradation of DMS was wavelength-dependent and the contributions of different wavebands to DMS photodegradation process were quite different. In order to assess the photodegradation process of DMS. . ." (Page 9 Line 17-19)

Question: The authors mention the following on page 3 lines 11-13: "Moreover, we examined the photolysis rate and concentration changes of DMS under the dual stressors of changing pH and solar radiation/temperature and assessed the coupling effects of OA, solar radiation and warming on biogenic dimethylated sulfur compounds cycling." Where is the basic information on light? How was light measured? How did it vary

naturally over the 23 day experiment? Where is the basic information on temperature? How was it measured? How did it vary over time? I do not see this anywhere. This is very concerning.

Reply: In this research, photodegradation experiments and temperature experiments were two separate experiments, both of which took 8 hours (Page 4 Line 21-22). The light intensity and UV intensity were measured using the agricultural environment detector (Zhejiang Top Instrument Co., Ltd.) and UVR meter (Beijing Normal University Optoelectronics Instrument Factory) and the temperature was detected using thermometer. The variations of light intensity and temperature during experiment were shown in the figure 2. (Page 23 Line 1-4) The relevant content was added in the revised manuscript: "The light intensity and UV intensity were measured using the agricultural environment detector (Zhejiang Top Instrument Co., Ltd.) and UVR meter (Beijing Normal University Optoelectronics Instrument Factory) and the temperature was detected using thermometer. The variations of pH, light intensity and temperature during experiments are shown in Fig. 2." (Page 4 Line 24-27)

Question: Furthermore, the authors seem to treat light and temperature as a single fixed factor (light/temperature) (? I may be wrong because the text is convoluted and unclear) and proceed with a two-way ANOVA to disentangle the impacts of pH, light, and temperature on concentrations of sulphur compounds? Ecologically and statistically, this is difficult to justify. pH + Light experiments were carried out under natural ambient solar radiation (lines 139-140) with 3 treatments and 1 control (full light) and then information is inferred for a 5th variable by subtracting the effects of one treatment from another treatment. The authors state that they also ran pH + Temperature experiments (with 1 treatment (+4oC) and control but it sounds like these are separate experiments from the light+pH experiments, however it is rather unclear): Lines 140-142: "Temperature experiments were also carried out with unfiltered water in quartz bottle under in situ temperature at 12 oC and high temperature at 18 oC for 8 h. The temperature was continuously controlled by circulating in situ seawater, hot water, and

ice." (This alone begs the question of the uniformity of the temperature treatment itself if ice and hot water are necessary to stabilize the temperature (there were likely a lot of fluctuations, yet these are not reported)). In any case, I have serious doubts as to the validity of lumping together Light and Temperature variables into a single fixed factor as it seems to be proposed in the statistical paragraph of the methodological section. Also are the underlying assumptions of normal distribution for the response variables (S compounds) respected? There is no mention of this. The authors state the following at lines 25-28 p.5: "Seawater pH was adjusted with CO2-saturated seawater during the experiment to maintain a stable pH environment. In the first week, pH level was comparatively stable with a better control, but pH fluctuated obviously and was difficult to control in the stable and decline phases of algal growth. As shown in Fig. 1, the pH showed relatively apparent fluctuations on days 9-12 and days 18-20." Again, it is difficult here to conceive that pH was a fixed explanatory variable. Judging by Figure 1, there are several instances when there doesn't appear to be a statistical difference between the pH treatments with very large error bars. On page 5 Lines 14-20, the authors describe very briefly the statistical approaches used, but these spark more questions than answer anything. "Statistical analysis was performed using SPSS version 18.0 (SPSS Inc., IBM, USA). Pearson's correlation coefficient and probability (p) were calculated to evaluate the quality of the fit when variables were normally distributed. (How was normality assessed?) T-test was used to determine whether a significant difference existed between two treatments. (WHICH treatment? More information is needed.) "Variability in the concentrations of biogenic dimethylated sulfur compounds was analyzed using two-way ANOVA with pH and temperature/light as fixed factors and concentration as a random factor to understand whether an interaction existed between pH and temperature/light on concentration. (Are the authors suggesting that temperature/light are one fixed factor?) Value at $p \leq 0.05$ was considered statistically significant. Linear correlation analyses were used to determine the response of DMS, DMSO concentrations, and bacterial abundance to OA using Origin 9.1." The authors proceed to establish Pearson's correlations between two variables within a same pH

condition. I am not convinced that this is useful. What is the purpose of this exactly? How does it inform us of the complex response of these variables to fluctuations of pH itself? Overall, the statistical schemes proposed by the authors do not seem appropriate, and they do not facilitate the interpretation of possibly complex responses. Table 4 presents these correlation coefficients. What are the degrees of freedom (DF) for these analyses? Why is a coefficient of 0.791 (between DMSOd and chla) significant at the p <0.05 level while a coefficient of 0.778 (between DMSOd and DMS) is significant at the p <0.01 level? More information is needed about the n and the DF. On pages 10 and 11, the authors discuss these correlations between sulphur compounds and other variables (within a same pH treatment) at length but I am not convinced that these are very informative on the impact of pH itself on the sulphur compounds. When looking at figure 2-3-4-5, the error bars are so wide for the pH treatments (substantial overlapping most of the time, ex: DMSOp), that it is hard to understand what the purpose of these inner-treatment correlations is. A very convoluted text does not help in the matter. An example of this here: "A significant correlation between DMS and DMSOd was observed in LC treatments (Table 4). The result was in accordance with the findings of Hatton et al. (2004) and Zindler-Schlundt et al. (2015) who showed that DMSO and DMS are closely related because of the direct formation of DMSO via bacterial DMS oxidation. The close link between DMS and DMSOd was not observed in MC and HC treatments. The changes in the production, consumption, and degradation processes which caused by the decreasing pH might mask the relationship between DMS and DMSOd. (Production/consumption and degradation of what exactly?) In addition, DMSPd concentration showed an obvious correlation with DMSOd only in LC treatments for the entire duration of the experiment (Table 4). DMSPd may be cleaved to DMS by specific bacteria that contain DMSP lyase (Curson et al., 2008), followed by the bacterial oxidation of DMS to DMSOd. This relationship was not found in MC and HC treatments. The phenomenon might be caused by the change in phytoplankton community and bacterial oxidation of DMS to DMSO." It is very difficult to follow the logic here. What change in phyto community? What are the authors talking about? They do

not show any information on community composition for this study. And this is another important shortcoming of this paper: no phytoplankton identification. DMSP/DMS cycling is intimately linked with species composition, yet we have no idea WHO is there and HOW pH may affect the primary producers responsible for a substantial part of the DMS/DMSP cycling. In the introduction the authors mention this: "Shaw hypothesized that DMS and sulfate aerosols are linked to global climate. This link was further elaborated by Charlson et al. (1987). Consequently, ocean acidification (OA)-induced changes in the primary productivity might impact on the production rate and sea-to-air emission of DMS and these impacts might further affect cloud formation and climate". The authors do not explore the primary production side of things at all. There are no PP rates, no phyto identification, only chla which is a proxy of biomass that does not inform us on whether pH impacts chla levels through physiological processes or through variability in the species composition.

Reply: For the statistical analyses, we used a one-sample Kolmogorov–Smirnov test and Levene test to confirm normal distribution of data and check for homogeneity, respectively. T-test was used to determine whether a significant difference existed between two treatments. Because the conditions during the incubation experiment showed fluctuations, we used the average pH value as the fixed factor and tried our best to estimate the impact of pH on DMS production and release. The analytical method was performed according to the method reported by Arnold et al. (2013). "Arnold, H. E., Kerrison, P., and Steinke, M.: Interacting effects of ocean acidification and warming on growth and DMS-production in the haptophyte coccolithophore Emiliania huxleyi, Global Change Biol., 19, 1007–1016, https://doi.org/10.1111/gcb.12105, 2013." "degree of freedom" refers to the number of items of data that are free to vary independently. When applied to a set of quantitative data, for a specified value of the mean, only (n-1) items are free to vary. A coefficient of 0.791 (between DMSOd and Chl-a) significant at the p <0.05 level means that DMSOd and Chl-a showed weak correlation in 95% confidence interval. While DMSOd and DMS showed a strong correlation (a coefficient of 0.778) even within the 99% confidence interval. This indicated

that DMSOd presented a good correlativity with DMS. DMSOd was more closely related to DMS than Chl-a. For Table 4, the value of n was 15. We have added this information in the revised manuscript. "Table 4. Pearson's correlation coefficient and associated significance level for mean dimethylated sulfur compounds and Chl-a concentrations under three different $CO_2$ treatments (n=15)." (Page 34 Line 2) Thanks for the reviewer's comment for sentences "The changes in the production, consumption, and degradation processes which caused by the decreasing pH might mask the relationship between DMS and DMSOd" and "The phenomenon might be caused by the change in phytoplankton community and bacterial oxidation of DMS to DMSO". We agree with the reviewer's comment and reworded this sentence in the revised manuscript. "The changes in the conversion of DMSOd to DMS caused by the decreasing pH might mask the relationship between DMS and DMSOd." (Page 11 Line 33- Page 12 Line 1) "This result might be because low pH changed the conversion process among DMSPd, DMS and DMSOd, which broke the relationship between DMSPd and DMSOd. This conclusion further confirmed the inference (section 4.1.1) that OA mainly influenced DMS concentration by affecting the transformation of dimethylated sulfur compounds." (Page 12 Line 4-7) In this research, phytoplankton community was determined only at the beginning and the end of the experiment. At the beginning of the experiment, the phytoplankton community included diatom, dinoflagellate, chrysophyceae, and cryptophyceae with a mean abundance of 2135 cells $L-1$ (Page 6 Line 13-14). At the end of the experiment, phytoplankton species were less in HC than in LC, and chrysophyceae (e.g., coccolith) and cryptophyceae (e.g., Cryptomonas sp. and Rhodomonas sp.) disappeared during HC treatment (Page 6 Line 23-24). Some previous literature (Zhang et al., 2008) reported that Chl-a was an indicator of phytoplankton biomass, so changes in the phytoplankton biomass were represented by Chl-a concentration during the experiment. "Zhang, H. H., Yang, G. P., Zhu, T.: Distribution and cycling of dimethylsulfide (DMS) and dimethylsulfoniopropionate (DMSP) in the sea-surface microlayer of the Yellow Sea, China, in spring, Contin. Shelf Res., 28, 2417–2427. https://doi.org/10.1016/j.csr.2008.06.003, 2008"

Question: 1.4 Out-dated information and statements that are too general. Page 1 line 28. The information given here is not up to date. The authors reference a paper written in 2000. Progress has been made in the last ca. 15 years. Anthropogenic carbon dioxide (CO2) emissions have increased atmospheric CO2 concentrations from their pre-industrial value of 280 to ca. 400 $\mu$atm in 2016 (NOAA-ESRL). Page 1 lines 29-30. Under which scenario exactly? Please be precise and use the latest information available. Concentrations of 850 to 1370 $\mu$atm are expected by the end of the century under the business-as-usual scenario RCP 8.5 (IPCC, 2013). Page 1 line 30 and beyond. The authors don't offer any explanation, even if short, as to the underlying processes involved in pH modulation in oceans as a result of CO2 increases in the atmosphere. They should spend a little more time explaining this. That is one the focuses of their paper after all. Also there is a more recent paper by Caldeira and Wickett (2005) suggesting that the surface ocean pH is expected to decrease by an additional 0.3-0.4 units under the RCP 8.5 scenario by 2100. Page 2 lines 1-2. What is the reference here? There are several consequences of increasing OA. Modifications in DIC is the primary, from which modifications in carbonate system ensue (which includes modifications in saturation states of calcite and aragonite, which by the way the authors don't explain the importance of: why talk about calcite and aragonite?). Modifications in the amount of protons (H+) is another. Etc. Etc. Page 2 lines 2-4. This is quite a general statement. What changes exactly (less calcite, more H+, more CO2?) will affect what physiological processes exactly, and what type of marine "organism" exactly? There is so much literature on the many aspects of OA and the consequences for marine organisms (Calcifiers? Phyto? Zoo? Fish? Etc), and the potential impacts may be positive, negative, resilience etc. . . The phrase written is simply too generalist, it should offer at least a glimpse into the multi-tiered discoveries made by the OA research community, or at least focus on the aspects that are relevant to the author's research: DMS-producing microbial communities. . . Page 2 lines 19-20. The authors make very general statements without offering much information: "The conversion of DMSP to DMS is controlled by a number of chemical and biological

processes." Yes. But which ones? These are only some examples of the generalities, more can be found in the manuscript.

Reply: Thanks for the reviewer's comment. We have rephrased the relevant content and updated new literature in the manuscript. "The current atmospheric CO2 concentration (pCO2) is around 400 $\mu$atm (NOAA-ESRL) and rising at an unprecedented rate. Concentrations of 850 to 1370 $\mu$atm are expected by the end of the century under the business-as-usual scenario RCP 8.5 (IPCC, 2013). The oceanic uptake of anthropogenic CO2 emissions is leading to a change of seawater carbonate system, manifested as increasing protons [H+], falling [CO32-] and a drop in seawater pH. Equilibration of seawater with increasing CO2 concentration in the atmosphere has already declined the surface ocean pH levels by 0.3–0.4 units compared to pre-industrial values (Caldeira and Wickett, 2005). The primary consequence of the inexorable increase in oceanic acidity is a change in seawater carbonate system. This change could affect marine ecological environment and primary productivity, as low pH will affect physiological process of marine phytoplankton (Fu et al., 2007; Mélancon et al., 2016; Orr et al., 2005). For example, pH conditions can influence the intracellular acid-base balance and the energy demand of phytoplankton (Kramer et al., 2003). Physiological activities of phytoplankton may be influenced, and therefore may affect the production and release of biogenic compounds, such as marine biogenic trace gas. The oceans are an important source of some atmosphere trace gases which affect atmospheric chemistry and global climate (Arnold et al., 2013; Asher et al., 2016; Park et al., 2014). Change in surface ocean pH will have an important impact on the climate." (Page 1 Line 28-30, Page 2 Line 1-10) "Caldeira, K., and Wickett, M. E.: Oceanography: anthropogenic carbon and ocean pH, Nature, 425, 365–365, https://doi.org/10.1038/425365a, 2003" has replaced with "Caldeira, K., and Wickett, M. E.: Ocean model predictions of chemistry changes from carbon dioxide emissions to the atmosphere and ocean, J. Geophys. Res. Oceans, 110, https://doi.org/10.1029/2004JC002671, 2005." (Page 15 Line 30-31) "The conversion of DMSP to DMS is controlled by a number of chemical and biological factors (Archer et al., 2002), such as activity of DMSP-lyase enzymes

(Stefels et al., 2000) and condition of light (Slezak et al., 2007)." (Page 2 Line 22-24) "Slezak, D., Kiene, R. P., Toole, D. A., Simó, R., Kieber, D. J.: Effects of solar radiation on the fate of dissolved DMSP and conversion to DMS in seawater, Aquat. Sci., 69, 377–393, https://doi.org/10.1007/s00027-007-0896-z, 2007." (Page 19 Line 28-29) "Stefels, J.: Physiological aspects of the production and conversion of DMSP in marine algae and higher plants, J. Sea Res. 43, 183–197, https://doi.org/10.1016/S1385-1101(00)00030-7, 2000." (Page 20 Line 7-8)

Question: 2. On the form As it currently stands, the overall level of language does not meet the high standards of Limnology and Oceanography There are countless examples of awkward formulations and missing words throughout the paper, including: ". . .more CO2 (low pH) will affect physiological process of marine organism." Page 2 line 3 Page 2 lines 5 and beyond: The transition between the impact of OA on they physiology of marine organisms and the oceanic production of climate-relevant gases is not well established.

Reply: In order to meet the high standards of Biogeosciences, we have modified the whole article and the sentence ". . .more CO2 (low pH) will affect physiological process of marine organism" has been replaced with ". . .low pH will affect physiological process of marine phytoplankton. . ." (Page 2 Line 5) "This change could affect marine ecological environment and primary productivity, as low pH will affect physiological process of marine phytoplankton (Fu et al., 2007; Mélancon et al., 2016; Orr et al., 2005). For example, pH conditions can influence the intracellular acid-base balance and the energy demand of phytoplankton (Kramer et al., 2003). Physiological activities of phytoplankton may be influenced, and therefore may influence the production and release of biogenic compounds, such as marine biogenic trace gas. The oceans are an important source of some atmosphere trace gases which affect atmospheric chemistry and global climate (Arnold et al., 2013; Asher et al., 2016; Park et al., 2014). Change in surface ocean pH will have an important impact on the climate." (Page 2 Line 4-10)

Question: Once emitted to atmosphere, DMS. . .". Page 2 line 10 "DMSO was initially

conceived as a sink for DMS. Nevertheless, DMSO was later found a potential source of DMS." Page 2 lines 26-27 "The ocean undergoes multiple environmental changes. Other climatic ecological stressor or factors would probably alter the effects of ocean acidification on the production and consumption process of DMS in both direct and indirect ways." Page 3 lines 2-4 "The photochemical process of DMS in the surface water would change due to the changing light level and seawater pH level." Page 3 line 5 There are so many more examples. I will stop here because I believe a profound language editing needs to be conducted and this goes beyond the scientific mandate of a reviewer. The introduction offers many general statements that only skim a fraction of the inherent complexity and wealth of information related to OA research. The paper lacks coherence and clarity and there is redundancy in the writing. Formulation of phrases is awkward, making the text very difficult to follow. Page 2 lines 6-8: Redundancy in the phrase: "It is produced by enzymatic cleavage of dimethylsulfonio-propionate (DMSP) (Gabric et al., 2010), which is synthesized by marine phytoplankton as a phytoplankton-derived precursor of DMS." Page 2 lines 16-17: Redundancy in the phrase as well as unclear: "DMSP, as the main precursor of DMS, . . ., presents an important effect on the biogeochemical cycle of climatically active trace gas DMS." This is rather vague and redundant. Needs rephrasing.

Reply: According to the reviewer's suggestion, we have revised the relevant sentences in the revised manuscript. The sentence "It is produced by enzymatic cleavage of dimethylsulfoniopropionate (DMSP) (Gabric et al., 2010), which is synthesized by marine phytoplankton as a phytoplankton-derived precursor of DMS." and the reference "Gabric, A. J., Cropp, R., Hirst, T., and Marchant, H.: The sensitivity of dimethyl sulfide production to simulated climate change in the Eastern Antarctic Southern Ocean, Tellus, 55, 966–981, https://doi.org/10.1034/j.1600-0889.2003.00077.x, 2010" have been deleted. The sentence "Nevertheless, DMSO was later found a potential source of DMS (Hatton et al., 2012). DMSO can be biologically reduced to DMS via enzymatic reactions that might depend on reductases (Spiese et al., 2009). Thus, DMSO is a key compound in the complex redox loop that is involved in marine sulfur cycle because it

can be both an end product of DMS oxidation and a precursor of DMS, which potentially plays an important role in climate regulation (Charlson et al., 1987; Deschaseaux et al., 2014)" has been replaced with "DMSO and DMS can be converted to each other through photochemical and biological oxidation processes (Brimblecombe and Shooter, 1986; Toole and Siegel, 2004; Hatton et al., 2012; Spiese et al., 2009). Thus, DMSO is a key compound in the complex redox loop that is involved in marine sulfur cycle, which potentially plays an important role in climate regulation (Charlson et al., 1987; Deschaseaux et al., 2014)." (Page 2 Line 30-33)

Question: Page 2 line 21 The word "preferred" is not appropriate.

Reply: We agree with the reviewer's comment and have deleted the "preferred and" in the manuscript.

Question: Page 2 line 26: "DMSO was originally conceived as a sink for DMS." This phrase does not make sense. Needs rewording. Again, these are only examples, more can be found in the manuscript. In short, at this point, there is much work to be done before this paper can be considered for publication in Limnology and Oceanography. First and foremost, there are several uncertainties and questions related to the methodologies used and the experimental design itself. Without a sound experimental plan and measurements, the rest of it (interpretations and conclusions) is useless. Overall, the entire methodological aspects related to the OA part of the experiments are unclear and lacking (carbonate system measurements absent, salinity?, temperature records?). It is unclear whether the temperature and light treatments were separate experiments. And where are those measurements of temperature and light? What about primary production, phytoplankton species composition?The statistical schemes do not seem appropriate. The objectives of the study are not well stated (I am referring here to the Estuarine context in which the experiment takes place). The environmental setting is not clearly established; the authors don't even situate the study area on a map. The level of language is not up to par.

Reply: According to the reviewer's suggestion, we have changed "DMSO was originally conceived as a sink for DMS" to "DMSO can also be directly synthesized by phytoplankton and plays similar biological functions to DMSP (Gao et al., 2017). DMSO and DMS can be converted to each other through photochemical and biological oxidation processes (Brimblecombe and Shooter, 1986; Toole and Siegel, 2004; Hatton et al., 2012; Spiese et al., 2009)" in the revised manuscript. (Page 2 Line 29-31) "Gao, N., Yang, G. P., Zhang, H. H., Liu, L.: Temporal and spatial variations of three dimethylated sulfur compounds in the Changjiang Estuary and its adjacent area during summer and winter, Environ. Chem., 14, 160–177, http://dx.doi.org/10.1071/EN16158, 2017." (Page 17 Line 1-3) Yes, the temperature and light treatments were separate experiments. In order to better understand, we explained it in the revised manuscript and added the graph of temperature and light intensity in Figure 2. "In the separate photodegradation experiment,..." (Page 4 Line 12) "Another separate experiment of temperature was also carried out with unfiltered water in quartz bottle under in situ temperature at 12 °C and high temperature at 18 °C for 8 h." (Page 4 Line 21-22) Unfortunately, we measured the initial temperature (12.33 °C), salinity (33.79‰, and carbonate parameters of three treatments only at the beginning of the experiment, the parameters during the experiments were not determined. Thanks for the reviewer's comments and we will monitor these parameters in real time in our future research. In order to make the study area more clearly, we have marked the sampling station on a map (see Figure 1). (Page 22 Line 1-2)
* * *
[Figure]

Figure 1. Location of the sampling station (123.50 °E, 30.56 °N) during the cruise.

**Fig. 1.**

[Figure]

Figure 2. Temporal changes in pH (a) for averages of the nine barrels over the triplicates treatments during ocean acidification incubation experiment, variations of light intensity (b) during photodegradation experiment and temperature (c) during temperature experiment.

**Fig. 2.**

---

## Author Comment (AC2) · 3 Feb 2018

Question 1. Lack of reference to the existing literature: There are now a good number of published papers out there that report the effects of OA on DMS and related compounds, but the authors have only cited a handful. In particular, the works of Archer et al. (Arctic mesocosm), Webb et al. (various mesocosms), and Hopkins et al. (various mesocosms and shipboard studies), Hussherr et al. (shipboard) are not mentioned at all, which seems a little odd. To me, it is important to place your findings within the context of the existing literature. They do mention a few examples, but with some errors

in referencing: e.g. Avgoustidi et al. 2012 is given in the reference list, but they also mention Avgoustidi 2006 in the text but not the list of references.

Reply: Thanks for the reviewer's comment. We have added these new literatures in the revised manuscript. "Other climatic ecological stressor or factors would probably alter the effects of ocean acidification on the production and consumption process of DMS in both direct and indirect ways (Arnold et al., 2013; Archer et al., 2013; Webb et al., 2016; Hopkins et al., 2010; Hussherr et al., 2017)." (Page 3 Line 10) "Archer, S. D., Kimmance, S. A., Stephens, J. A., Hopkins, F. E., Bellerby, R. G. J., Schulz, K. G., Piontek, J., Engel, A.: Contrasting responses of DMS and DMSP to ocean acidification in Arctic waters, Biogeosciences, 10, 1893–1908, https://doi.org/10.5194/bg-10-1893-2013, 2013." (Page 14 Line 30-32) "Webb, A. L., Malin, G., Hopkins, F. E., Ho, K. L., Riebesell, U., Schulz, K. G., Larsen, A., Liss, P. S.: Ocean acidification has different effects on the production of dimethylsulfide and dimethylsulfoniopropionate measured in cultures of Emiliania huxleyi and a mesocosm study: a comparison of laboratory monocultures and community interactions, Environ. Chem, 35, 405–420, https://doi.org/10.1071/EN14268, 2016." (Page 20 Line 20-23) "Hopkins, F. E., Turner, S. M., Nightinale, P. D., Steinke, M., Liss, P. S.: Ocean acidification and marine biogenic trace gas emissions, P. Natl. Acad. Sci. USA., 107, 760–765. https://doi.org/10.1073/pnas.0907163107, 2010." (Page 17 Line 18-19) "Hussherr, R., Levasseur, M., Lizotte, M., Tremblay, J., Mol, J., Thomas, H., Gosselin, M., Starr, M., Miller, L. A., Jarniková,T., Schuback, N., Mucci, A.: Impact of ocean acidification on arctic phytoplankton blooms and dimethyl sulfide concentration under simulated ice-free and under-ice conditions, Biogeosciences, 14, 2407-2427, https://doi.org/10.5194/bg-14-2407-2017, 2017." (Page 17 Line 20-23) "Avgoustidi (2006)" has been replaced with "Avgoustidi (2012)" in Page 10 Line 27.

Question 2. Lack of appreciation of the 'bigger picture': Why do we perform these kinds of experiments? Ultimately, it is to generate data that may be utilised by modellers within earth system models. Some recognition/discussion of this would be useful. At

least, some discussion of published studies (Six et al., Schwinger et al.) to provide the reader with an understanding of the potential DMS-climate feedbacks resulting from OA. Otherwise why do we care?

Reply: Thanks for the reviewer's comment. We agree with the reviewer's comment and have added the relevant literature in the manuscript. "Six et al. (2013) and Schwinger et al. (2017) estimated changes in future DMS emissions with Earth system model and indicated that global warming can be amplified by reduced production as a result of OA. Therefore, the research on biogeochemical cycle of DMS can help to better understand the feedback effect between the OA and global warming. Moreover,..." (Page 3 Line 3-6) "Six, K. D., Kloster, S., Ilyina, T., Archer, S. D., Zhang, K., and Maier-Reimer, E.: Global warming amplified by reduced sulphur fluxes as a result of ocean acidification, Nature Clim. Change, 3, 975–978, https://doi.org/10.1038/nclimate1981, 2013." (Page 19 Line 25-27) "Schwinger, J., Tjiputra, J., Goris, N., Six, K. D., Kirkevåg, A., Seland, Ø., Heinze, C., Ilyina, T.: Amplification of global warming through pH dependence of DMS production simulated with a fully coupled Earth system model, Biogeosciences, 14, 3633–3648, https://doi.org/10.5194/bg-14-3633-2017, 2017." (Page 19 Line 17-19)

Question 3. Language weaknesses: The entire paper needs language checking. The structure of the paper needs some consideration - for example, large chunks of the 'Results' text would probably be considered more suitable for the 'Discussion'. I would recommend that the authors carefully consider all of the above and the detailed points raised by Reviewer #1 before even considering resubmitting this paper. In its current form, it is far from suitable for publication.

Reply: Thanks for the reviewer's comment. We have adjusted the structure of the 'Results' and 'Discussion'. "The concentrations of DMSPp showed significant differences among three treatments ($p \leq 0.008$) for high-temperature treatments and minimal differences under ambient temperature treatments. Results from two-way ANOVA illustrated the interaction between temperature and pH on DMSPp concentration ($p = 0.001$). This indicated that the pH and temperature influenced the biological production of the dimethylated sulfur compounds in seawater." has been pulled together into: "4.2 The effect of the interaction between OA and environmental factors". The revised content appears in Page 13 Line 17-26: "The concentrations of DMSPp showed significant differences among three treatments ($p \leq 0.008$) for high-temperature treatments and minimal differences under ambient temperature treatments. Results from two-way ANOVA illustrated the interaction between temperature and pH on DMSPp concentration ($p = 0.001$). This result indicated that temperature could regulate the effect of pH on the biological productions of DMS and DMSPp in seawater. In addition, Todgham and Stillman (2013) reported that multiple stressors could influence performance independently (additive) or interactively (antagonistic or synergistic). We can infer that OA and warming play a synergistic role in the production of dimethylated sulfur compounds. This result is consistent with that reported by Six et al. (2013) who showed that global warming could be amplified through pH dependence of DMS production. Therefore, global environmental change manifesting in OA and warming may not result in a decreased DMS as suggested by the effect of elevated $CO_2$ in isolation" In order to improve the level of English, the manuscript has been edited for language by EnPapers: http://www.enpapers.com/

In short, we have carefully considered the reviewers' comments and suggestions and conducted the revision seriously. We are very thankful to the reviewers for all the valuable comments and helpful suggestions to improve this manuscript.

[Figure]

**EnPapers.Com**                    **CERTIFICATE OF ENGLISH EDITING**

To whom it may concern:

This memo is to certify that the paper titled ______Effects of the interaction of ocean acidification,solar radiation,and warming on biogenic dimethylated sulfur compounds cycling in the Changjiang River Estuary___ has been edited for language by EnPapers, a company dedicated to helping international researchers publish their findings in the best English language journals possible.

Our International paper editing service is performed by a subject expert editor and approved by two senior editors. All our editors are native English speakers.

The certificate is being issued upon the request of the client. If you have any questions, please contact papers@enpapers.com

Signature of the editor representative:

Martin J. Booth

**Fig. 1.** CERTIFICATE_OF_ENGLISH_EDITING

---

## Author Comment (AC3) · 4 Feb 2018

Dear Editor and Reviewers, Thank you for your useful comments and suggestions on the language and structure of our manuscript (bg-2017-453). The manuscript has been carefully revised according to reviewers' comments and polished by the polishing company. The following are the reviewer's comments related to the manuscript and how we have addressed each of reviewer's concerns (red words). Changes have been marked as red in the manuscript.

Question 1. On the core. 1.1 Questionable methodological approach The methodological approach used by the authors is incomplete and raises several concerns. The most alarming one is trying to report the impacts of OA on the dynamics of biogenic sulphur compounds and having only this to say about carbonate system measurements and monitoring throughout the 23-day experiment: p.3 Lines 29-30 "Seawater pH was constantly measured using a pH meter (. . .) and the precision is +/- 0.002." Have the authors taken into account a correction factor based on fluctuating temperature? pH meters are usually calibrated at 25oC and substantial variation in measurements can ensue from variability in temperature. The authors mention that ". . .temperature was continuously controlled by circulating in situ seawater, hot water, and ice.", suggesting that potentially significant variation in temperature occurred during 23 days. Have they monitored salinity throughout the experiment? The pH scale (National Bureau of Standard scale (pHNBS), free scale (pHF), total scale (pHT), or seawater scale (pHSWS)) is not mentioned in the manuscript.

Reply: Thanks for the reviewer's comment. Yes, we have taken into account a correction factor based on fluctuating temperature. pH meters were calibrated at 25 oC in the determination and three parallel samples were measured for each pH value. Temperature was continuously controlled by circulating in situ seawater, hot water, and ice in the temperature experiments for 8 h (not the incubation experiment of ocean acidification for 23 days). In the incubation experiment of ocean acidification (23 days), all incubation barrels were exposed to sunlight in a flow-through water bath to maintain treatments at their in situ temperature. Unfortunately, the salinity data were not determined throughout the experiment. According to the reviewer's suggestion, we have added the relevant content in the revised manuscript. "Seawater pH was continually measured using a pH meter (S210 SevenCompact™, Mettler, Germany, precision: ± 0.002), using the National Bureau of Standards (NBS) scale to calibrate the pH electrode." (Page 4 Line 9-11)

Question: To ensure reproducibility, it is critical to report and monitor at least two variables of the carbonate system of seawater (as well as salinity and temperature) for the entire period of the experiment. The authors do add a separate paragraph at the end of the Analytical procedures section where they mention the following: p.5 Lines 9-12 "Total dissolved inorganic carbon (DIC) was determined by a DIC analyzer (AS-C2, Apollo SciTech Inc., Georgia, USA). 10 A sample of 0.5 mL was acidified by 0.5 mL 10

Reply: Thanks for the reviewer's comment. Unfortunately, no temporal monitoring is available in our experiment. DIC was measured only at the beginning of the incubation experiment and the Standard Deviations have been added in Table 1 in (Page 31 Line 1).

A volume of 0.5 mL is sufficient for the measurement of DIC. We have added references in the revised manuscript. "Cai, W. J., Dai, M., Wang, Y., Zhai, W., Huang, T., Chen, S., Zhang, F., Chen, Z., Wang, Z.: The biogeochemistry of inorganic carbon and nutrients in the Pearl River estuary and the adjacent Northern South China Sea. Contin. Shelf Res., 24, 1301–1319, https://doi.org/10.1016/j.csr.2004.04.005, 2004" (Page 15 Line 25-27) "Cai, W. J., Wang, Y.: The chemistry, fluxes, and sources of carbon dioxide in the estuarine waters of the Satilla and Altamaha Rivers, Georgia, Limnol. Oceanogr., 43, 657–668, https://doi.org/10.4319/lo.1998.43.4.0657, 1998." (Page 15 Line 28-29)

Question: Furthermore, the authors do not mention the formulations used to calculate all variables with CO2SYS: concentrations of total boron, $CO_2$ solubility (K0), Dissociation constants of carbonic acid (K1 and K2), boric acid (Kb), water (Kw), phosphoric acid (Kp1, Kp2, Kp3), silicic acid (Ksi), hydrogen fluoride (Kf), and bisulfate (Ks), Solubility products of calcite (Kspc) and aragonite (Kspa). A very useful document prepared in the framework of the data management activity of the Ocean Acidification International Coordination Centre of the International Atomic Energy Agency can be found here (OAICCL: www.iaea.org/ocean-acidification) and shares recommendations proposed in the Guidelines for reporting ocean acidification data in scientific journals. I also worry about trying to measure concentrations of volatile DMS within non collapsible barrels from which volumes of water are extracted daily creating more and more

headspace over the course of 23 days (figure 3). The authors do not offer any explanation for this, or any suggestions as to how they correct for this problem. We learn that the authors used filtered (what porosity? Axenic? 0.2um, 0.7um?) water to run side-incubations to measure the impact of light on DMS. A first thought here is that any type of filtration process may introduce biases. DMSP-producing communities are notoriously sensitive to filtration, the most common problem is the pressureinduced heightening of the DMSPd pool caused by the rupture of healthy cells. This can artificially enhance pools of DMS through mixing of DMSPd and DMSP-lyase enzymes. This is never mentioned. Also, DMS is controlled by both biology (bacteria, viruses, phyto, zoo) and physics (light, wind). Thus the response of DMS to various light regimes may occur through various pathways (photochemically but also through its impact on primary producers or bacterioplankton). The methodological scheme suggests that the authors here are only investigating the impact of pH and light on photolysis rates of DMS. Yet it is unclear if bacteria/phyto are still present in these experiments (what is the porosity of the filter used?). On page 8 line 25 the authors state the following: "In order to assess the community-level response to the ocean change in future $CO_2$ and light conditions, photolysis rate constants (K, d−1) for DMS were calculated." How is it possible to assess a "community-level" response when filtered water is used presumably to focus on physical processes only (photochemistry)? The authors state objectives that cannot be answered through the methods they use.

Reply: Thanks for the reviewer's comment. We agree with the reviewer's comment. CO2SYS: The value of K0 (the solubility coefficient of $CO_2$) and the conversion between the fugacity and the partial pressure of $CO_2$ are from Weiss, R. F., Marine Chemistry 2:203-215, 1974. The value of Kb (for boric acid), in constant choices 1 to 5, is from Dickson, Andrew G., Deep-Sea Research 37:755-766, 1990. The boron concentration in constant choices 1 to 5 is from Uppstrom, Leif, Deep-Sea Research 21:161-162, 1974. Values of KW (for H2O), KP1, KP2, and KP3 (for phosphoric acid), and KSi (for silicic acid) are from (in constant choices 1 to 5) Millero, Frank J., Geochemica et Cosmochemica Acta 59:661-677, 1995. K1 and K2 of carbonic acid are from the data

that refit by Dickson and Millero (1987) on the seawater scale. In the present study, we measured the DMS concentration in solution after the gas-liquid equilibrium. Changes in DMS production by phytoplankton will result in a simultaneous change in DMS concentration in the solution after the equilibration between solution and headspace, so the presence of the headspace will not affect the overall trend of DMS concentration changes. The seawater using in the photodegradation experiment was filtered with 0.4 and 0.2 $\mu$m Whatman GF/F filter and all equipment were sterilized during the experiment. In addition, the DMS samples were collected and measured without any filtration process. We have added the relevant content in the revised manuscript. "In the separate photodegradation experiment, 1 L seawater sample was filtered directly into precleaned Qorpak bottles through 0.45 and 0.2 $\mu$m polyethersulfone membranes to remove bacteria, phytoplankton, zooplankton and original DMS. Then the standard solution of DMS was added to the filtered seawater." (Page 4 Line 12-14) Furthermore, we have rephrased the sentence "In order to assess the community-level response to the ocean change in future CO2 and light conditions, photolysis rate constants (K, d−1) for DMS were calculated" in the revised manuscript. "In order to assess the photodegradation process of DMS response to the ocean change in future CO2 and light conditions, photolysis rate constants (K, d−1) for DMS were calculated." (Page 9 Line 18-19)

Question: The authors use 20L barrels to run their 23 day incubation experiment. Of those 23 days we learn that 18 are onboard the ship and 5 are conducted in the lab (page 3 line 16)? This is very worrisome. No further information is given here. How were the barrels moved? Presumably a lot of mixing occurred during the transit? How were temperatures kept constant during this time? For an experiment looking to identify the impacts of light and temperature, (and pH) on the dynamics of a DMSP-DMS-producing community, this major shift in environmental settings introduces a lot of potential variability that can obscure the response. Yet this is simply glanced over and never mentioned again. Also, I am curious to know (this could be calculated from the methodologies) how much water was left in the 20L barrels prior to the last sampling

day? At least 10L? Or less?

Reply: The barrels were moved by a truck. The barrels were covered with lids throughout the experiment, including the transportation process, which can avoid the gas loss caused by mixing. The relevant content has been added in the manuscript. "All barrels were sealed with lids throughout the experiment, including the transportation process". (Page 4 Line 6-7) The transport took about twelve hours, during which time the temperature was controlled by the seawater in the incubation tank to keep the incubation barrels at the same temperature. By calculating, approximately 11.6 L seawater was left in the 20 L barrels on the last sampling day.

Question: 1.2 The authors treat the response of a complex Estuary to OA the same way they would the response of open waters to OA. The OA problem is increasingly complex in estuarine and coastal waters where freshwater runoff, tidal mixing and high biological activity contribute to variations in CO2 and pH on different time scales. The surface mixed-layer pCO2 can vary spatially and is strongly modulated by biological productivity during the phytoplankton growth season. Surface pHT in Estuaries can also vary significantly within a single tidal cycle, nearly as much as the world's ocean have experienced in response to anthropogenic CO2 uptake over the last century. . . Studying the impact of OA in these circumstances (high natural variability in pH and possible resilience of communities) is not devoid of interest, but the authors do not state this, they don't even mention these natural fluctuations and how these could affect the communities, on a basic level, are these communities already tolerant to rapid fluctuations in pH? They treat the impact of OA on this very complex ecosystem as if it was the impact of OA on open oceanic waters. This is troublesome.

Reply: Thanks for the reviewer's comment. We agree with the reviewer's comment and have added relevant content in the revised manuscript. "In recent years, the effects of simulated future CO2-induced seawater acidification on DMS production are still controversial (Avgoustidi et al., 2012; Kerrison et al., 2012; Kim et al., 2010; Vogt et al., 2008) and very few data are available on the OA effects on DMS production in

estuaries. The patterns of low pH states observed in complex estuary are largely due to the result of freshwater runoff, natural mixing and high biological activity. By the end of this century, OA may become the dominant process reducing the pH in estuary area (Feely et al., 2010). Studying the impact of OA on the DMS cycle in estuary is very necessary and meaningful (Reum et al., 2015)." (Page 3 Line 15-20). "Feely, R. A., Alin, S. R., Newton, J., Sabine, C. L., Warner, M., Devol, A., Krembs, C., Maloy, C.: The combined effects of ocean acidification, mixing, and respiration on pH and carbonate saturation in an urbanized estuary. Estuar. Coast. Shelf Sci., 88, 442–449, https://doi.org/10.1016/j.ecss.2010.05.004, 2010." (Page 16 Line 21-23) In the acidification incubation experiment, chrysophyceae and cryptophyceae disappeared during HC treatment at the late stage of the experiment, indicating a great influence of OA on their growth. But they made up a smaller percentage of this community and diatoms and dinoflagellates dominated the phytoplankton community. According the result of section 4.1.3, the ratios of DMS/Chl-a, DMSPt/Chl-a, and DMSOt/Chl-a showed no difference in the dimethylated sulfur compounds production in terms of phytoplankton biomass. This result indicated that the relative strength of DMS, DMSP, and DMSO production by the phytoplankton communities has not changed and these communities were tolerant to rapid fluctuations in pH. In order to better understand, we added relevant content in the revised manuscript. "This result indicated that the productions of dimethylated sulfur compounds per unit biomass were not affected by pH perturbations and these communities were tolerant to rapid fluctuations in pH." (Page 12 Line 29-31)

Question: 1.3 Statistical approach The description of the statistical strategies used by the authors is confusing and their application to the dataset is questionable. The title of the paper is full of promise "Effects of the interaction of OA, solar radiation and warming on biogenic sulphur compounds cycling. . ." but fails to deliver on that promise. The methodological and statistical approach proposed by the authors does not allow them to explore potential interactive effects of three stressors on the sources and sinks of DMS and DMSP. Multiple stressors can influence a variable independently (additive), or interact to either reduce (antagonistic) or enhance (synergistic) that variable in a nonlinear, unpredictable fashion. I suggest the authors read Todgham and Stillman (2013), Riebesell and Gattuso (2015), Reum et al. (2015), Gunderson et al (2016). A collapsed factorial approach would have been more informative and simpler to interpret, see Boyd et al (2015). Please also see: "SCOR working group 149 https://scor149-ocean.com Changing Ocean Biological Systems (COBS): How will biota respond to a changing ocean?" It contains pertinent information for moving studies from single to multiple drivers.

Reply: In this study, we explored the potential interactive effects of ocean acidification and solar radiation or warming, the title of the paper may be improperly expressed. Thanks for the reviewer's comment and we have changed the title and related narration in the revised manuscript. "Effects of the interaction of ocean acidification and solar radiation/warming on biogenic dimethylated sulfur compounds cycling in the Changjiang River Estuary" (Page 1 Line 1-3) "Moreover, we examined the photolysis rate and concentration changes of DMS under the dual stressors of changing pH and solar radiation/temperature and assessed the coupling effects of OA and solar radiation/warming on biogenic dimethylated sulfur compounds cycling." (Page 3 Line 22-23) "3.2 DMS and DMSP concentration response to the interaction among OA and solar radiation/warming" (Page 9 Line 15) Thanks to the references recommended by the reviewer. We have added these references and relevant interpretation in the revised manuscript. "So these concomitant global change variables should be taken into consideration to better explain the effect of acidification on DMS (Todgham and Stillman, 2013; Boyd et al., 2015; Riebesell and Gattuso, 2015; Gunderson et al., 2016)." (Page 3 Line 14) "Studying the impact of OA on the DMS cycle in estuary is very necessary and meaningful (Reum et al., 2015)" (Page 3 Line 19-20) "Todgham, A. E., Stillman, J. H.: Physiological responses to shifts in multiple environmental stressors: relevance in a changing world, Integr. Comp. Biol., 53, 539–544, https://doi.org/10.1093/icb/ict086, 2013." (Page 20 Line 9-10) "Boyd P W, Lennartz S T, Glover D M, Doney, S. C.: Biological ramifications of climate-change-mediated oceanic multi-stressors, Nat. Clim. Change, 5, 71–79, https://doi.org/10.1038/nclimate2441, 2015." (Page 15 Line 18-

19) "Riebesell, U., Gattuso, J. P.: Lessons learned from ocean acidification research, Nat. Clim. Change, 5, 12–14, https://doi.org/10.1038/nclimate2456, 2015." (Page 19 Line 13-14) "Gunderson, A. R., Armstrong, E. J., Stillman, J. H.: Multiple stressors in a changing world: the need for an improved perspective on physiological responses to the dynamic marine environment, Ann. Rev. Mar. Sci., 8, 357–378, https://doi.org/10.1146/annurev-marine-122414-033953, 2016." (Page 17 Line 6-8) "Reum J C P, Alin S R, Harvey C J, Bednaršek, N., Evans, W., Feely, R. A., Hales, B., Lucey, N., Mathis, J. T., McElhany, P., Newton, J. Sabine, C. L.: Interpretation and design of ocean acidification experiments in upwelling systems in the context of carbonate chemistry co-variation with temperature and oxygen. ICES J Mar. Sci., 73, 582–595, https://doi.org/10.1093/icesjms/fsu231, 2015." (Page 19 Line 9-12) "In addition, Todgham and Stillman (2013) reported that multiple stressors can influence performance independently (additive) or interact interactively (antagonistic or synergistic). We can infer that OA and warming play a synergistic role in the production of dimethylated sulfur compounds. This result is consistent with that reported by Six et al. (2013) who showed that global warming can be amplified through pH dependence of DMS production." (Page 13 Line 21-25)

Question: On page 12 lines 24-26 the authors mention the following: "The effect of the interaction between OA and environmental conditions complicates the overall ecosystem response. Hence, comprehensive consideration of OA and solar radiation can better interpret and understand feedbacks between OA and global climatic change." A very small exploration of interactive effects is shown in Figure 6 for pH and temperature, and there appears to be a subset of information on pH and certain measurements of light in Table 3, but the interactive impacts of solar radiation, pH and temperature together are not explored. I do not see any "comprehensive consideration". Table 3 seems to relate one type of pH with one type of light treatment, but no combinations. . . pH 8.1 + light control (quartz), pH 8.1 + Mylar D (UVA), pH8.1 + Plexiglass (PAR), pH8.1 + Dark. pH + light control (quartz), pH 8.1 + Mylar D (UVA), pH8.1 + Plexiglass (PAR), pH8.1 + Dark. Same with pH 7.9 or 7.7 with various light treatments. It is not

clear what the rational is behind choosing a specific pH with a specific light treatment (?).

Reply: According to the reviewer's suggestion, we have reworded this sentence in the revised manuscript. "Hence, consideration of the interaction between OA and solar radiation can better interpret and understand feedbacks between OA and global climatic change." (Page 13 Line 14-15) Deal et al. (2005) reported that the contributions of UVA, UVB and PAR to DMS photodegradation were quite different and Bouillon and Miller (2005) mentioned that pH level can influence the photodegradation of DMS. So the contribution of UVA, UVB and PAR to DMS photodegradation under different pH levels may different. Furthermore, the ocean undergoes ocean acidification. To better understand the DMS loss through photodegradation in the future ocean, we designed this experiment with five light conditions (natural light, UVA, visible light, dark, UVB: results for full-spectrum natural light minus those for UVB-filtered light) and three pH levels (ambient level: 8.1; level expected at the end of this century: 7.9; level predicted for the middle of the next century: 7.7). "The research of Deal et al. (2005) mentioned that the photodegradation of DMS was wavelength-dependent and the contributions of different wavebands to DMS photodegradation process were quite different. In order to assess the photodegradation process of DMS. . ." (Page 9 Line 17-19)

Question: The authors mention the following on page 3 lines 11-13: "Moreover, we examined the photolysis rate and concentration changes of DMS under the dual stressors of changing pH and solar radiation/temperature and assessed the coupling effects of OA, solar radiation and warming on biogenic dimethylated sulfur compounds cycling." Where is the basic information on light? How was light measured? How did it vary naturally over the 23 day experiment? Where is the basic information on temperature? How was it measured? How did it vary over time? I do not see this anywhere. This is very concerning.

Reply: In this research, photodegradation experiments and temperature experiments were two separate experiments, both of which took 8 hours (Page 4 Line 21-22). The

light intensity and UV intensity were measured using the agricultural environment detector (Zhejiang Top Instrument Co., Ltd.)  and UVR meter (Beijing Normal University Optoelectronics Instrument Factory) and the temperature was detected using thermometer.  The variations of light intensity and temperature during experiment were shown in the figure 2.  (Page 23 Line 1-4) The relevant content was added in the revised manuscript: "The light intensity and UV intensity were measured using the agricultural environment detector (Zhejiang Top Instrument Co., Ltd.) and UVR meter (Beijing Normal University Optoelectronics Instrument Factory) and the temperature was detected using thermometer. The variations of pH, light intensity and temperature during experiments are shown in Fig. 2." (Page 4 Line 24-27)

Question: Furthermore, the authors seem to treat light and temperature as a single fixed factor (light/temperature) (? I may be wrong because the text is convoluted and unclear) and proceed with a two-way ANOVA to disentangle the impacts of pH, light, and temperature on concentrations of sulphur compounds? Ecologically and statistically, this is difficult to justify.  pH + Light experiments were carried out under natural ambient solar radiation (lines 139-140) with 3 treatments and 1 control (full light) and then information is inferred for a 5th variable by subtracting the effects of one treatment from another treatment. The authors state that they also ran pH + Temperature experiments (with 1 treatment (+4oC) and control but it sounds like these are separate experiments from the light+pH experiments, however it is rather unclear): Lines 140-142: "Temperature experiments were also carried out with unfiltered water in quartz bottle under in situ temperature at 12 oC and high temperature at 18 oC for 8 h. The temperature was continuously controlled by circulating in situ seawater, hot water, and ice." (This alone begs the question of the uniformity of the temperature treatment itself if ice and hot water are necessary to stabilize the temperature (there were likely a lot of fluctuations, yet these are not reported)). In any case, I have serious doubts as to the validity of lumping together Light and Temperature variables into a single fixed factor as it seems to be proposed in the statistical paragraph of the methodological section. Also are the underlying assumptions of normal distribution for the response variables

(S compounds) respected? There is no mention of this. The authors state the following at lines 25-28 p.5: "Seawater pH was adjusted with CO2-saturated seawater during the experiment to maintain a stable pH environment.  In the first week, pH level was comparatively stable with a better control, but pH fluctuated obviously and was difficult to control in the stable and decline phases of algal growth. As shown in Fig. 1, the pH showed relatively apparent fluctuations on days 9-12 and days 18-20." Again, it is difficult here to conceive that pH was a fixed explanatory variable. Judging by Figure 1, there are several instances when there doesn't appear to be a statistical difference between the pH treatments with very large error bars. On page 5 Lines 14-20, the authors describe very briefly the statistical approaches used, but these spark more questions than answer anything.  "Statistical analysis was performed using SPSS version 18.0 (SPSS Inc., IBM, USA). Pearson's correlation coefficient and probability (p) were calculated to evaluate the quality of the fit when variables were normally distributed. (How was normality assessed?)  T-test was used to determine whether a significant difference existed between two treatments. (WHICH treatment? More information is needed.) "Variability in the concentrations of biogenic dimethylated sulfur compounds was analyzed using two-way ANOVA with pH and temperature/light as fixed factors and concentration as a random factor to understand whether an interaction existed between pH and temperature/light on concentration. (Are the authors suggesting that temperature/light are one fixed factor?) Value at $p \leq 0.05$ was considered statistically significant. Linear correlation analyses were used to determine the response of DMS, DMSO concentrations, and bacterial abundance to OA using Origin 9.1." The authors proceed to establish Pearson's correlations between two variables within a same pH condition. I am not convinced that this is useful. What is the purpose of this exactly? How does it inform us of the complex response of these variables to fluctuations of pH itself? Overall, the statistical schemes proposed by the authors do not seem appropriate, and they do not facilitate the interpretation of possibly complex responses. Table 4 presents these correlation coefficients. What are the degrees of freedom (DF) for these analyses? Why is a coefficient of 0.791 (between DMSOd and chla) significant at the

p <0.05 level while a coefficient of 0.778 (between DMSOd and DMS) is significant at the p <0.01 level? More information is needed about the n and the DF. On pages 10 and 11, the authors discuss these correlations between sulphur compounds and other variables (within a same pH treatment) at length but I am not convinced that these are very informative on the impact of pH itself on the sulphur compounds. When looking at figure 2-3-4-5, the error bars are so wide for the pH treatments (substantial overlapping most of the time, ex: DMSOp), that it is hard to understand what the purpose of these inner-treatment correlations is. A very convoluted text does not help in the matter. An example of this here: "A significant correlation between DMS and DMSOd was observed in LC treatments (Table 4). The result was in accordance with the findings of Hatton et al. (2004) and Zindler-Schlundt et al. (2015) who showed that DMSO and DMS are closely related because of the direct formation of DMSO via bacterial DMS oxidation. The close link between DMS and DMSOd was not observed in MC and HC treatments. The changes in the production, consumption, and degradation processes which caused by the decreasing pH might mask the relationship between DMS and DMSOd. (Production/consumption and degradation of what exactly?) In addition, DMSPd concentration showed an obvious correlation with DMSOd only in LC treatments for the entire duration of the experiment (Table 4). DMSPd may be cleaved to DMS by specific bacteria that contain DMSP lyase (Curson et al., 2008), followed by the bacterial oxidation of DMS to DMSOd. This relationship was not found in MC and HC treatments. The phenomenon might be caused by the change in phytoplankton community and bacterial oxidation of DMS to DMSO." It is very difficult to follow the logic here. What change in phyto community? What are the authors talking about? They do not show any information on community composition for this study. And this is another important shortcoming of this paper: no phytoplankton identification. DMSP/DMS cycling is intimately linked with species composition, yet we have no idea WHO is there and HOW pH may affect the primary producers responsible for a substantial part of the DMS/DMSP cycling. In the introduction the authors mention this: "Shaw hypothesized that DMS and sulfate aerosols are linked to global climate. This link was further

elaborated by Charlson et al. (1987). Consequently, ocean acidification (OA)-induced changes in the primary productivity might impact on the production rate and sea-to-air emission of DMS and these impacts might further affect cloud formation and climate". The authors do not explore the primary production side of things at all. There are no PP rates, no phyto identification, only chla which is a proxy of biomass that does not inform us on whether pH impacts chla levels through physiological processes or through variability in the species composition.

Reply: For the statistical analyses, we used a one-sample Kolmogorov–Smirnov test and Levene test to confirm normal distribution of data and check for homogeneity, respectively. T-test was used to determine whether a significant difference existed between two treatments. Because the conditions during the incubation experiment showed fluctuations, we used the average pH value as the fixed factor and tried our best to estimate the impact of pH on DMS production and release. The analytical method was performed according to the method reported by Arnold et al. (2013). "Arnold, H. E., Kerrison, P., and Steinke, M.: Interacting effects of ocean acidification and warming on growth and DMS-production in the haptophyte coccolithophore Emiliania huxleyi, Global Change Biol., 19, 1007–1016, https://doi.org/10.1111/gcb.12105, 2013." "degree of freedom" refers to the number of items of data that are free to vary independently. When applied to a set of quantitative data, for a specified value of the mean, only (n-1) items are free to vary. A coefficient of 0.791 (between DMSOd and Chl-a) significant at the p <0.05 level means that DMSOd and Chl-a showed weak correlation in 95For Table 4, the value of n was 15. We have added this information in the revised manuscript. "Table 4. Pearson's correlation coefficient and associated significance level for mean dimethylated sulfur compounds and Chl-a concentrations under three different CO2 treatments (n=15)." (Page 34 Line 2) Thanks for the reviewer's comment for sentences "The changes in the production, consumption, and degradation processes which caused by the decreasing pH might mask the relationship between DMS and DMSOd" and "The phenomenon might be caused by the change in phytoplankton community and bacterial oxidation of DMS to DMSO". We agree with

the reviewer's comment and reworded this sentence in the revised manuscript. "The changes in the conversion of DMSOd to DMS caused by the decreasing pH might mask the relationship between DMS and DMSOd." (Page 11 Line 33- Page 12 Line 1) "This result might be because low pH changed the conversion process among DMSPd, DMS and DMSOd, which broke the relationship between DMSPd and DMSOd. This conclusion further confirmed the inference (section 4.1.1) that OA mainly influenced DMS concentration by affecting the transformation of dimethylated sulfur compounds." (Page 12 Line 4-7) In this research, phytoplankton community was determined only at the beginning and the end of the experiment. At the beginning of the experiment, the phytoplankton community included diatom, dinoflagellate, chrysophyceae, and cryptophyceae with a mean abundance of 2135 cells $L-1$ (Page 6 Line 13-14). At the end of the experiment, phytoplankton species were less in HC than in LC, and chrysophyceae (e.g., coccolith) and cryptophyceae (e.g., Cryptomonas sp. and Rhodomonas sp.) disappeared during HC treatment (Page 6 Line 23-24). Some previous literature (Zhang et al., 2008) reported that Chl-a was an indicator of phytoplankton biomass, so changes in the phytoplankton biomass were represented by Chl-a concentration during the experiment. "Zhang, H. H., Yang, G. P., Zhu, T.: Distribution and cycling of dimethylsulfide (DMS) and dimethylsulfoniopropionate (DMSP) in the sea-surface microlayer of the Yellow Sea, China, in spring, Contin. Shelf Res., 28, 2417–2427. https://doi.org/10.1016/j.csr.2008.06.003, 2008"

Question: 1.4 Out-dated information and statements that are too general. Page 1 line 28. The information given here is not up to date. The authors reference a paper written in 2000. Progress has been made in the last ca. 15 years. Anthropogenic carbon dioxide (CO2) emissions have increased atmospheric CO2 concentrations from their pre-industrial value of 280 to ca. 400 $\mu$atm in 2016 (NOAA-ESRL). Page 1 lines 29-30. Under which scenario exactly? Please be precise and use the latest information available. Concentrations of 850 to 1370 $\mu$atm are expected by the end of the century under the business-as-usual scenario RCP 8.5 (IPCC, 2013). Page 1 line 30 and beyond. The authors don't offer any explanation, even if short, as to the underlying

processes involved in pH modulation in oceans as a result of CO2 increases in the atmosphere. They should spend a little more time explaining this. That is one the focuses of their paper after all. Also there is a more recent paper by Caldeira and Wickett (2005) suggesting that the surface ocean pH is expected to decrease by an additional 0.3-0.4 units under the RCP 8.5 scenario by 2100. Page 2 lines 1-2. What is the reference here? There are several consequences of increasing OA. Modifications in DIC is the primary, from which modifications in carbonate system ensue (which includes modifications in saturation states of calcite and aragonite, which by the way the authors don't explain the importance of: why talk about calcite and aragonite?). Modifications in the amount of protons (H+) is another. Etc. Etc. Page 2 lines 2-4. This is quite a general statement. What changes exactly (less calcite, more H+, more CO2?) will affect what physiological processes exactly, and what type of marine "organism" exactly? There is so much literature on the many aspects of OA and the consequences for marine organisms (Calcifiers? Phyto? Zoo? Fish? Etc), and the potential impacts may be positive, negative, resilience etc. . . The phrase written is simply too generalist, it should offer at least a glimpse into the multi-tiered discoveries made by the OA research community, or at least focus on the aspects that are relevant to the author's research: DMS-producing microbial communities. . . Page 2 lines 19-20. The authors make very general statements without offering much information: "The conversion of DMSP to DMS is controlled by a number of chemical and biological processes." Yes. But which ones? These are only some examples of the generalities, more can be found in the manuscript.

Reply: Thanks for the reviewer's comment. We have rephrased the relevant content and updated new literature in the manuscript. "The current atmospheric CO2 concentration (pCO2) is around 400 $\mu$atm (NOAA-ESRL) and rising at an unprecedented rate. Concentrations of 850 to 1370 $\mu$atm are expected by the end of the century under the business-as-usual scenario RCP 8.5 (IPCC, 2013). The oceanic uptake of anthropogenic CO2 emissions is leading to a change of seawater carbonate system, manifested as increasing protons [H+], falling [CO32-] and a drop in seawater pH. Equilibration of seawater with increasing CO2 concentration in the atmosphere has already declined the surface ocean pH levels by 0.3–0.4 units compared to pre-industrial values (Caldeira and Wickett, 2005). The primary consequence of the inexorable increase in oceanic acidity is a change in seawater carbonate system. This change could affect marine ecological environment and primary productivity, as low pH will affect physiological process of marine phytoplankton (Fu et al., 2007; Mélancon et al., 2016; Orr et al., 2005). For example, pH conditions can influence the intracellular acid-base balance and the energy demand of phytoplankton (Kramer et al., 2003). Physiological activities of phytoplankton may be influenced, and therefore may affect the production and release of biogenic compounds, such as marine biogenic trace gas. The oceans are an important source of some atmosphere trace gases which affect atmospheric chemistry and global climate (Arnold et al., 2013; Asher et al., 2016; Park et al., 2014). Change in surface ocean pH will have an important impact on the climate." (Page 1 Line 28-30, Page 2 Line 1-10) "Caldeira, K., and Wickett, M. E.: Oceanography: anthropogenic carbon and ocean pH, Nature, 425, 365–365, https://doi.org/10.1038/425365a, 2003" has replaced with "Caldeira, K., and Wickett, M. E.: Ocean model predictions of chemistry changes from carbon dioxide emissions to the atmosphere and ocean, J. Geophys. Res. Oceans, 110, https://doi.org/10.1029/2004JC002671, 2005." (Page 15 Line 30-31) "The conversion of DMSP to DMS is controlled by a number of chemical and biological factors (Archer et al., 2002), such as activity of DMSP-lyase enzymes (Stefels et al., 2000) and condition of light (Slezak et al., 2007)." (Page 2 Line 22-24) "Slezak, D., Kiene, R. P., Toole, D. A., Simó, R., Kieber, D. J.: Effects of solar radiation on the fate of dissolved DMSP and conversion to DMS in seawater, Aquat. Sci., 69, 377–393, https://doi.org/10.1007/s00027-007-0896-z, 2007." (Page 19 Line 28-29) "Stefels, J.: Physiological aspects of the production and conversion of DMSP in marine algae and higher plants, J. Sea Res. 43, 183–197, https://doi.org/10.1016/S1385-1101(00)00030-7, 2000." (Page 20 Line 7-8)

Question: 2. On the form As it currently stands, the overall level of language does not meet the high standards of Limnology and Oceanography There are countless

examples of awkward formulations and missing words throughout the paper, including: ". . .more CO2 (low pH) will affect physiological process of marine organism." Page 2 line 3 Page 2 lines 5 and beyond: The transition between the impact of OA on they physiology of marine organisms and the oceanic production of climate-relevant gases is not well established.

Reply: In order to meet the high standards of Biogeosciences, we have modified the whole article and the sentence ". . .more CO2 (low pH) will affect physiological process of marine organism" has been replaced with ". . .low pH will affect physiological process of marine phytoplankton. . ." (Page 2 Line 5) "This change could affect marine ecological environment and primary productivity, as low pH will affect physiological process of marine phytoplankton (Fu et al., 2007; Mélancon et al., 2016; Orr et al., 2005). For example, pH conditions can influence the intracellular acid-base balance and the energy demand of phytoplankton (Kramer et al., 2003). Physiological activities of phytoplankton may be influenced, and therefore may influence the production and release of biogenic compounds, such as marine biogenic trace gas. The oceans are an important source of some atmosphere trace gases which affect atmospheric chemistry and global climate (Arnold et al., 2013; Asher et al., 2016; Park et al., 2014). Change in surface ocean pH will have an important impact on the climate." (Page 2 Line 4-10)

Question: Once emitted to atmosphere, DMS. . .". Page 2 line 10 "DMSO was initially conceived as a sink for DMS. Nevertheless, DMSO was later found a potential source of DMS." Page 2 lines 26-27 "The ocean undergoes multiple environmental changes. Other climatic ecological stressor or factors would probably alter the effects of ocean acidification on the production and consumption process of DMS in both direct and indirect ways." Page 3 lines 2-4 "The photochemical process of DMS in the surface water would change due to the changing light level and seawater pH level." Page 3 line 5 There are so many more examples. I will stop here because I believe a profound language editing needs to be conducted and this goes beyond the scientific mandate of a reviewer. The introduction offers many general statements that only skim a fraction of the inherent complexity and wealth of information related to OA research. The paper lacks coherence and clarity and there is redundancy in the writing. Formulation of phrases is awkward, making the text very difficult to follow. Page 2 lines 6-8: Redundancy in the phrase: "It is produced by enzymatic cleavage of dimethylsulfoniopropionate (DMSP) (Gabric et al., 2010), which is synthesized by marine phytoplankton as a phytoplankton-derived precursor of DMS." Page 2 lines 16-17: Redundancy in the phrase as well as unclear: "DMSP, as the main precursor of DMS, . . ., presents an important effect on the biogeochemical cycle of climatically active trace gas DMS." This is rather vague and redundant. Needs rephrasing.

Reply: According to the reviewer's suggestion, we have revised the relevant sentences in the revised manuscript. The sentence "It is produced by enzymatic cleavage of dimethylsulfoniopropionate (DMSP) (Gabric et al., 2010), which is synthesized by marine phytoplankton as a phytoplankton-derived precursor of DMS." and the reference "Gabric, A. J., Cropp, R., Hirst, T., and Marchant, H.: The sensitivity of dimethyl sulfide production to simulated climate change in the Eastern Antarctic Southern Ocean, Tellus, 55, 966–981, https://doi.org/10.1034/j.1600-0889.2003.00077.x, 2010" have been deleted. The sentence "Nevertheless, DMSO was later found a potential source of DMS (Hatton et al., 2012). DMSO can be biologically reduced to DMS via enzymatic reactions that might depend on reductases (Spiese et al., 2009). Thus, DMSO is a key compound in the complex redox loop that is involved in marine sulfur cycle because it can be both an end product of DMS oxidation and a precursor of DMS, which potentially plays an important role in climate regulation (Charlson et al., 1987; Deschaseaux et al., 2014)" has been replaced with "DMSO and DMS can be converted to each other through photochemical and biological oxidation processes (Brimblecombe and Shooter, 1986; Toole and Siegel, 2004; Hatton et al., 2012; Spiese et al., 2009). Thus, DMSO is a key compound in the complex redox loop that is involved in marine sulfur cycle, which potentially plays an important role in climate regulation (Charlson et al., 1987; Deschaseaux et al., 2014)." (Page 2 Line 30-33)

Question: Page 2 line 21 The word "preferred" is not appropriate.

Reply: We agree with the reviewer's comment and have deleted the "preferred and" in the manuscript.

Question: Page 2 line 26: "DMSO was originally conceived as a sink for DMS." This phrase does not make sense. Needs rewording. Again, these are only examples, more can be found in the manuscript. In short, at this point, there is much work to be done before this paper can be considered for publication in Limnology and Oceanography. First and foremost, there are several uncertainties and questions related to the methodologies used and the experimental design itself. Without a sound experimental plan and measurements, the rest of it (interpretations and conclusions) is useless. Overall, the entire methodological aspects related to the OA part of the experiments are unclear and lacking (carbonate system measurements absent, salinity?, temperature records?). It is unclear whether the temperature and light treatments were separate experiments. And where are those measurements of temperature and light? What about primary production, phytoplankton species composition?The statistical schemes do not seem appropriate. The objectives of the study are not well stated (I am referring here to the Estuarine context in which the experiment takes place). The environmental setting is not clearly established; the authors don't even situate the study area on a map. The level of language is not up to par.

Reply: According to the reviewer's suggestion, we have changed "DMSO was originally conceived as a sink for DMS" to "DMSO can also be directly synthesized by phytoplankton and plays similar biological functions to DMSP (Gao et al., 2017). DMSO and DMS can be converted to each other through photochemical and biological oxidation processes (Brimblecombe and Shooter, 1986; Toole and Siegel, 2004; Hatton et al., 2012; Spiese et al., 2009)" in the revised manuscript. (Page 2 Line 29-31) "Gao, N., Yang, G. P., Zhang, H. H., Liu, L.: Temporal and spatial variations of three dimethylated sulfur compounds in the Changjiang Estuary and its adjacent area during summer and winter, Environ. Chem., 14, 160–177, http://dx.doi.org/10.1071/EN16158,

2017." (Page 17 Line 1-3) Yes, the temperature and light treatments were separate experiments. In order to better understand, we explained it in the revised manuscript and added the graph of temperature and light intensity in Figure 2. "In the separate photodegradation experiment,..." (Page 4 Line 12) "Another separate experiment of temperature was also carried out with unfiltered water in quartz bottle under in situ temperature at 12 °C and high temperature at 18 °C for 8 h." (Page 4 Line 21-22) Unfortunately, we measured the initial temperature (12.33 °C), salinity (33.79‰, and carbonate parameters of three treatments only at the beginning of the experiment, the parameters during the experiments were not determined. Thanks for the reviewer's comments and we will monitor these parameters in real time in our future research. In order to make the study area more clearly, we have marked the sampling station on a map (see Figure 1). (Page 22 Line 1-2)

Please also note the supplement to this comment:
https://www.biogeosciences-discuss.net/bg-2017-453/bg-2017-453-AC3-supplement.pdf
* * *
[Figure]

Figure 1. Location of the sampling station (123.50 °E, 30.56 °N) during the cruise.

**Fig. 1.** Figure 1

[Figure]

Figure 2. Temporal changes in pH (a) for averages of the nine barrels over the triplicates treatments during ocean acidification incubation experiment, variations of light intensity (b) during photodegradation experiment and temperature (c) during temperature experiment.

**Fig. 2.** Figure 2

Table 1. Preliminary carbonate parameters of three treatments during the incubation experiment.

| Treatments | pH ± s.d. | DIC ± s.d. (μmol kg$^{-1}$) | pCO$_2$ ± s.d. (μatm) | HCO$_3^-$ ± s.d. (μmol kg$^{-1}$) | CO$_3^{2-}$ ± s.d. (μmol kg$^{-1}$) |
|---|---|---|---|---|---|
| M1-3 | 8.1 ± 0.008 | 2271 ± 0.577 | 389.2 ± 5.661 | 2087.2 ± 3.371 | 168.0 ± 3.926 |
| M4-6 | 7.9 ± 0.012 | 2263 ± 1.000 | 635.9 ± 13.80 | 2130.0 ± 4.133 | 107.1 ± 4.093 |
| M7-9 | 7.7 ± 0.032 | 2274 ± 1.154 | 1040.7 ± 54.05 | 2164.0 ± 4.193 | 67.5 ± 7.552 |

**Fig. 3.** Table 1

**Supplement:**

**Effects of the interaction of ocean acidification and solar radiation/warming on biogenic dimethylated sulfur compounds cycling in the Changjiang River Estuary**

Shan Jian[1], Jing Zhang[1, 3], Hong-Hai Zhang[1, 3], and Gui-Peng Yang[1, 2, 3]

[1]Key Laboratory of Marine Chemistry Theory and Technology, Ocean University of China, Ministry of Education, Qingdao 266100, China

[2]Laboratory for Marine Ecology and Environmental Science, Qingdao National Laboratory for Marine Science and Technology, Qingdao 266071, China

[3]Institute of Marine Chemistry, Ocean University of China, Qingdao 266100, China

*Correspondence to*: Gui-Peng Yang (gpyang@ouc.edu.cn)

**Abstract.** Ocean acidification (OA) affects marine primary productivity and community structure, and therefore may influence the biogeochemical cycles of volatile biogenic dimethyl sulfide (DMS) and its precursor dimethylsulfoniopropionate (DMSP) and photochemical oxidation product dimethyl sulfoxide (DMSO). A 23-day incubation experiment on board was conducted to investigate the short-term response of biogenic sulfur compounds production and cycling to OA in the Changjiang River Estuary and further understand its effects on biogenic sulfur compounds. Result showed that phytoplankton abundance and species presented remarkable differences under three different pH levels in the late stage of the experiment. A significant reduction in chlorophyll *a* (Chl-*a*), DMS, particulate DMSP (DMSPp), and dissolved DMSO (DMSOd) concentrations was identified under high $CO_2$ levels. Moreover, minimal change was observed in the production of dissolved DMSP (DMSPd) and particulate DMSO (DMSOp) among treatments. The ratios of DMS, total DMSP (DMSPt), and total DMSO (DMSOt) to Chl-*a* were also not affected by a change in pH. In addition, DMS and DMSOd were highly related to mean bacterial abundance under three pH levels. Additional incubation experiments on light and temperature showed that the influence of pH on productions of dimethylated sulfur compounds also depended on solar radiation and temperature conditions. DMS photodegradation rate increased with decreasing pH under full-spectrum natural light and UVB light. Thus, OA may lead to decreasing DMS concentrations in the surface seawater. Light and temperature conditions also play an important role in the production and cycling of biogenic sulfur compounds.

Keywords: Dimethylated sulfur compounds, Ocean acidification, Solar radiation, Warming, Bacteria, Phytoplankton

**1 Introduction**

The current atmospheric $CO_2$ concentration ($pCO_2$) is around 400 μatm (NOAA-ESRL) and rising at an unprecedented rate. Concentrations of 850 to 1370 μatm are expected by the end of the century under the business-as-usual scenario RCP 8.5 (IPCC, 2013). The oceanic uptake of anthropogenic $CO_2$ emissions is leading to a change of seawater carbonate system,

manifested as increasing protons [H$^+$], falling [CO$_3^{2-}$] and a drop in seawater pH. Equilibration of seawater with increasing CO$_2$ concentration in the atmosphere has already declined the surface ocean pH levels by 0.3–0.4 units compared to pre-industrial values (Caldeira and Wickett, 2005). The primary consequence of the inexorable increase in oceanic acidity is a change in seawater carbonate system. This change could affect marine ecological environment and primary productivity, as low pH will affect physiological process of marine phytoplankton (Fu et al., 2007; Mdancon et al., 2016; Orr et al., 2005). For example, pH conditions can influence the intracellular acid-base balance and the energy demand of phytoplankton (Kramer et al., 2003). Physiological activities of phytoplankton may be influenced, and therefore may affect the production and release of biogenic compounds, such as marine biogenic trace gas. The oceans are an important source of some atmosphere trace gases which affect atmospheric chemistry and global climate (Arnold et al., 2013; Asher et al., 2016; Park et al., 2014). Change in surface ocean pH will have an important impact on the climate.

Dimethyl sulfide (DMS) is an important climatically active biogenic gas. It is the dominant volatile sulfur compound in ocean surface waters (Quinn and Bates, 2011) and its emissions from the ocean to the atmosphere corresponds to almost 90% of the marine biogenic sulfur in the atmosphere (Wakeham, 1986). Once emitted to atmosphere, DMS undergoes rapid oxidation to produce particles which, through direct and indirect interactions with incoming solar radiation, affect planetary albedo (Barnes et al., 2006; Charlson et al., 1987; Rap et al., 2013). Shaw hypothesized that DMS and sulfate aerosols are linked to global climate. This link was further elaborated by Charlson et al. (1987). Consequently, ocean acidification (OA)-induced changes in the primary productivity might impact on the production rate and sea-to-air emission of DMS and these impacts might further affect cloud formation and climate.

DMSP, as the main precursor of DMS, is produced by various phytoplankton species and presents an important effect on the biogeochemical cycle of climatically active trace gas DMS (Keller et al., 1989). DMSP is released from phytoplankton cells into the dissolved phase through active exudation or when cells are lysed during grazing (Wolfe et al., 2000), viral attack (Malin et al., 1998), or senescence (Stefels et al., 2007). The conversion of DMSP to DMS is controlled by a number of chemical and biological factors (Archer et al., 2002), such as activity of DMSP-lyase enzymes (Stefels et al., 2000) and condition of light (Slezak et al., 2007). DMSP is degraded in seawater via two main pathways. DMSP-lyase pathway contributes only a considerably small fraction to DMSP metabolism in seawater. The most dominant process is the dimethylation/demethiolation pathway, which diverts sulfur away from DMS production (Kiene and Bates, 1990). However, whether shifts in DMS yield occur under OA condition is unknown.

Dimethyl sulfoxide (DMSO) is the major nonvolatile dimethyl sulfur pool in seawater and also plays an important role in biogeochemical cycle of DMS. DMSO can also be directly synthesized by phytoplankton and plays similar biological functions to DMSP (Gao et al., 2017). DMSO and DMS can be converted to each other through photochemical and biological oxidation processes (Brimblecombe and Shooter, 1986; Toole and Siegel, 2004; Hatton et al., 2012; Spiese et al., 2009). Thus, DMSO is a key compound in the complex redox loop that is involved in marine sulfur cycle, which potentially plays an important role in climate regulation (Charlson et al., 1987; Deschaseaux et al., 2014).

Along with ocean acidification, global warming associated with the increasing atmospheric $CO_2$ accumulation is leading to a greenhouse ocean, which is characterized by increased sea surface temperature and a shoaling of the upper mixed layer (Capotondi et al., 2012; Doney, 2006; Gao et al., 2012a; Hays et al., 2005). Six et al. (2013) and Schwinger et al. (2017) estimated changes in future DMS emissions with Earth system model and indicated that global warming can be amplified by reduced production as a result of OA. Therefore, the research on biogeochemical cycle of DMS can help to better understand the feedback effect between the OA and global warming. Moreover, the declining stratospheric ozone over the globe and change in the upper mixed layer would enhance the levels of ultraviolet radiation (UVR) (280 to 400 nm) reaching the sea surface. The ocean undergoes multiple environmental changes. Other climatic ecological stressor or factors would probably alter the effects of ocean acidification on the production and consumption process of DMS in both direct and indirect ways (Arnold et al., 2013; Archer et al., 2013; Webb et al., 2016; Hopkins et al., 2010; Hussherr et al., 2017). A recent study predicts that pH level can influence the photodegradation of DMS (Bouillon and Miller, 2005). The photochemical process of DMS in the surface water would change due to the changing light level and seawater pH level. So these concomitant global change variables should be taken into consideration to better explain the effect of acidification on DMS (Todgham and Stillman, 2013; Boyd et al., 2015; Riebesell and Gattuso, 2015; Gunderson et al., 2016).

In recent years, the effects of simulated future $CO_2$-induced seawater acidification on DMS production are still controversial (Avgoustidi et al., 2012; Kerrison et al., 2012; Kim et al., 2010; Vogt et al., 2008) and very few data are available on the OA effects on DMS production in estuaries. The patterns of low pH states observed in complex estuary are largely due to the result of freshwater runoff, natural mixing and high biological activity. By the end of this century, OA may become the dominant process reducing the pH in estuary area (Feely et al., 2010). Studying the impact of OA on the DMS cycle in estuary is very necessary and meaningful (Reum et al., 2015). Hence, we conducted a shipboard incubation experiment in the Changjiang River Estuary to further investigate the OA effects in the cycling of marine DMS, DMSP, and DMSO under three pH treatments. Moreover, we examined the photolysis rate and concentration changes of DMS under the dual stressors of changing pH and solar radiation/temperature and assessed the coupling effects of OA and solar radiation/warming on biogenic dimethylated sulfur compounds cycling.

**2 Materials and methods**

**2.1 Experimental design**

[revised manuscript text omitted]

**3.2.1 Solar radiation**

The research of Deal et al. (2005) mentioned that the photodegradation of DMS was wavelength-dependent and the contributions of different wavebands to DMS photodegradation process were quite different. In order to assess the photodegradation process of DMS response to the ocean change in future $CO_2$ and light conditions, photolysis rate constants ($K$, $d^{-1}$) for DMS were calculated. The result showed that the natural logarithm of DMS concentrations had a good linear relationship with time, so the photodegradation of DMS follows a pseudo-first-order kinetics (Bouillon and Miller, 2005; Brugger et al., 1998). $K$ can be calculated by the slope of the regression line of natural logarithm of DMS concentrations against time. DMS loss was observed under full-spectrum natural light, UVB, UVA, visible light, and dark conditions in each pH treatment. DMS photolysis rate constants were significantly different in various conditions (Table 3). Furthermore, no significant loss in DMS was observed in the dark during control experiments. The rate constants of DMS photolysis were in the range of 4.02–6.32, 1.13–4.73, 1.35–2.37, and 0.24–0.52 $d^{-1}$ under full-spectrum natural light, UVB, UVA, and visible light with average values of 5.18, 3.02, 1.75, and 0.41 $d^{-1}$, respectively (Table 3). The rate constants of DMS photolysis under full-spectrum natural light were higher than those under UVB, UVA, and visible light. The UVB made an important contribution to DMS photodegradation. The rate constants of DMS photolysis increased with decreasing pH levels under full-spectrum natural light and UVB. However, a complete and contrasting result was obtained under UVA and visible light. The contributions to the total photolysis of UVB under decreasing pH conditions were increased from 28.0% to 74.9%. On the contrary, the contributions of UVA and visible light were decreased from 59.1% to 21.3% and from 12.9% to 3.83%,

[revised manuscript text omitted]

**4.2 The effect of the interaction between OA and environmental factors**

The photolysis rate constants of DMS were strongly affected by solar radiation and OA. The net effects of OA on DMS production were largely dependent on light conditions. Relatively, significant change was observed in $K_{UVB}$ and $K_{UVA}$, particularly $K_{UVB}$ increased by four times at pH 7.7 compared at pH 8.1, indicating that reduced pH levels accelerated the photodegradation of DMS under UVB conditions. The result was mainly because photodegradation of DMS was related to inducement of some oxidants (such as singlet oxygen, hydroxyl radical, hydrogen peroxide, and photoactivated CDOM) (Kieber et al., 1996; Brimblecombe and Shooter, 1986) and some ions (such as $HCO_3^-$, $CO_3^{2-}$, $NO_3^-$, and $Br^-$) (Bouillon and Miller, 2005). The short wavelength UVB played a decisive role in the formation and reaction of these oxidants and pH simultaneously influenced the free-radical production/scavenging processes involving ions, which may be responsible for the variation in photodegradation rate under different sunlight and pH conditions. Although the interaction of acidification and solar radiation is considerably complex, an overall decrease of pH in UVB would result in an eventual decrease in DMS, indicating that ocean acidification can promote DMS photooxidation under UVB. OA is not proceeding in isolation (Gao et al., 2012). The effect of the interaction between OA and environmental conditions complicates the overall ecosystem response. Hence, consideration of the interaction between OA and solar radiation can better interpret and understand feedbacks between OA and global climatic change.

Figure 7 shows the DMS and DMSPp concentration variabilities under increased $CO_2$ and temperature. The highest DMS and DMSPp concentrations were observed at 18 ℃ under pH 7.9 and pH 7.7, respectively. The concentrations of DMSPp showed significant differences among three treatments ($p \leq 0.008$) for high-temperature treatments and minimal differences under ambient temperature treatments. Results from two-way ANOVA illustrated the interaction between temperature and pH on DMSPp concentration ($p = 0.001$). This result indicated that temperature could regulate the effect of pH on the biological productions of DMS and DMSPp in seawater. In addition, Todgham and Stillman (2013) reported that multiple stressors could influence performance independently (additive) or interactively (antagonistic or synergistic). We can infer that OA and warming play a synergistic role in the production of dimethylated sulfur compounds. This result is consistent with that reported by Six et al. (2013) who showed that global warming could be amplified through pH dependence of DMS production. Therefore, global environmental change manifesting in OA and warming may not result in a decreased DMS as suggested by the effect of elevated $CO_2$ in isolation.

Experimental results can only be hypothesized because information on specific classification of DMS-utilizing bacteria in the experiment is not available. Although such variability in DMS, DMSP, and DMSO concentrations existed during the short-term incubation experiment, whether the differences can be attributed to the effect of pH still remains unclear. Considering the complex interactive effects of environmental factors, an accurate model with appropriate parameterization of the environmental factors should be developed, and such model can improve our understanding of the earth system's response to predicted global environmental change.

**5 Conclusion**

[revised manuscript text omitted]

Reum J C P, Alin S R, Harvey C J, Bednaršek, N., Evans, W., Feely, R. A., Hales, B., Lucey, N., Mathis, J. T., McElhany, P.,
10  Newton, J. Sabine, C. L.: Interpretation and design of ocean acidification experiments in upwelling systems in the context of carbonate chemistry co-variation with temperature and oxygen. ICES J. Mar. Sci., 73, 582–595, https://doi.org/10.1093/icesjms/fsu231, 2015.

Riebesell, U., Gattuso, J. P.: Lessons learned from ocean acidification research, Nat. Clim. Change, 5, 12–14, https://doi.org/10.1038/nclimate2456, 2015.

15  Rost, B., Riebesell, U., Burkhardt, S., and Sültemeyer, D.: Carbon acquisition of bloom-forming marine phytoplankton, Limnol. Oceanogr., 48, 55–67, https://doi.org/10.4319/lo.2003.48.1.0055, 2003.

Schwinger, J., Tjiputra, J., Goris, N., Six, K. D., Kirkevåg, A., Seland, Ø., Heinze, C., Ilyina, T.: Amplification of global warming through pH dependence of DMS production simulated with a fully coupled Earth system model, Biogeosciences, 14, 3633–3648, https://doi.org/10.5194/bg-14-3633-2017, 2017.

20  Shaw, G. E.: Bio-controlled thermostasis involving the sulfur cycle, Clim. Change, 5, 297–303, https://doi.org/10.1007/BF02423524, 1983.

Simó, R., Archer, S. D., Gilpin, L., and Stelfox-Widdicombe, C. E.: Coupled dynamics of dimethylsulfoniopropionate and dimethylsulfide cycling and the microbial food web in surface waters of the North Atlantic, Limnol. Oceanogr., 47, 53–61, https://doi.org/10.4319/lo.2002.47.1.0053, 2002.

25  Six, K. D., Kloster, S., Ilyina, T., Archer, S. D., Zhang, K., and Maier-Reimer, E.: Global warming amplified by reduced sulphur fluxes as a result of ocean acidification, Nature Clim. Change, 3, 975–978, https://doi.org/10.1038/nclimate1981, 2013.

Slezak, D., Kiene, R. P., Toole, D. A., Simó, R., Kieber, D. J.: Effects of solar radiation on the fate of dissolved DMSP and conversion to DMS in seawater, Aquat. Sci., 69, 377–393, https://doi.org/10.1007/s00027-007-0896-z, 2007.

30  Smith, D. C., Steward, G. F., Long, R. A., and Azam, F.: Bacterial mediation of carbon fluxes during A diatom bloom in A mesocosm, Deep Sea Res. Part II Top. Stud. Oceanogr., 42, 75–97, https://doi.org/10.1016/0967-0645(95)00005-B, 1995.

Spiese, C. E., Kieber, D. J., Nomura, C. T., and Kiene, R. P.: Reduction of dimethylsulfoxide to dimethylsulfide by marine phytoplankton, Limnol. Oceanogr., 54, 560–570, https://doi.org/10.4319/lo.2009.54.2.0560, 2009.

Spilling, K., Kai, G. S., Paul, A. J., Boxhammer, T., Achterberg, E. P., Hornick, T., Lischka, S., Stuhr, A., Bermúdez, R., Czerny, J., Crawfurd, K., Brussaard, C. P. D., Grossart, H. P., and Riebesell, U.: Effects of ocean acidification on pelagic carbon fluxes in a mesocosm experiment, Biogeosciences, 13, 1–30, https://doi.org/10.5194/bg-2016-56, 2016.

Stefels, J., Steinke, M., Turner, S., Malin, G., and Belviso, S.: Environmental constraints on the production and removal of the climatically active gas dimethylsulphide (DMS) and implications for ecosystem modelling, Biogeochemistry, 83, 245–275, https://doi.org/10.1007/s10533-007-9091-5, 2007.

Stefels, J.: Physiological aspects of the production and conversion of DMSP in marine algae and higher plants, J. Sea Res. 43, 183–197, https://doi.org/10.1016/S1385-1101(00)00030-7, 2000.

Todgham, A. E., Stillman, J. H.: Physiological responses to shifts in multiple environmental stressors: relevance in a changing world, Integr. Comp. Biol., 53, 539–544, https://doi.org/10.1093/icb/ict086, 2013.

Toole, D. A., and Siegel, D. A.: Light-driven cycling of dimethylsulfide (DMS) in the Sargasso Sea: Closing the loop, Geophys. Res. Lett., 31, 111–142, https://doi.org/10.1029/2004GL019581, 2004.

Toole, D. A., Siegel, D. A. and Doney, S. C.: A light-driven, one dimensional dimethylsulfide biogeochemical cycling model for the Sargasso Sea, J. Geophys. Res. Biogeosci., 113, 385–393, https://doi.org/10.1029/2007JG000426, 2008.

Vogt, M., Steinke, M., Turner, S., Paulino, A., Meyerhöfer, M., Riebesell, U., and Liss, P.: Dynamics of dimethylsulphoniopropionate and dimethylsulphide under different $CO_2$ concentrations during a mesocosm experiment, Biogeosciences, 5, 407–419, https://doi.org/10.5194/bg-5-407-2008, 2008.

Wakeham, S. G.: Oceanic dimethylsulfide: production during zooplankton grazing on phytoplankton, Science, 233, 1314–1316, https://doi.org/10.1126/science.233.4770.1314, 1986.

Webb, A. L., Malin, G., Hopkins, F. E., Ho, K. L., Riebesell, U., Schulz, K. G., Larsen, A., Liss, P. S.: Ocean acidification has different effects on the production of dimethylsulfide and dimethylsulfoniopropionate measured in cultures of Emiliania huxleyi and a mesocosm study: a comparison of laboratory monocultures and community interactions, Environ. Chem., 35, 405–420, https://doi.org/10.1071/EN14268, 2016.

Wingenter, O. W., Haase, K. B., Zeigler, M., Blake, D. R., Rowland, F. S., Sive, B. C., Paulino, A., Thyrhaug, R., Larsen, A., Schulz, K., Meyerhöfer, M., and Riebesell, U.: Unexpected consequences of increasing $CO_2$ and ocean acidity on marine production of DMS and $CH_2ClI$: Potential climate impacts, Geophys. Res. Lett., 34, 223–224, https://doi.org/10.1029/2006GL028139, 2007.

Wolfe, G. V., and Steinke, M.: Grazing-activated production of dimethyl sulfide (DMS) by two clones of Emiliania huxleyi, Limnol. Oceanogr., 41, 1151–1160, https://doi.org/10.4319/lo.1996.41.6.1151, 1996.

Wolfe, G. V., Levasseur, M., Cantin, G., and Michaud, S.: DMSP and DMS dynamics and microzooplankton grazing in the Labrador Sea: application of the dilution technique, Deep Sea Res., Part I, 47, 2243–2264, https://doi.org/10.1016/S0967-0637(00)00028-5, 2000.

Wu, Y., Gao, K. and Riebesell, U.: $CO_2$-induced seawater acidification affects physiological performance of the marine diatom Phaeodactylum tricornutum, Biogeosciences, 7, 2915–2923, https://doi.org/10.5194/bg-7-2915-2010, 2010.

Yoch, D. C., Ansede, J. H., and Rabinowitz, K. S.: Evidence for intracellular and extracellular dimethylsulfoniopropionate (DMSP) lyases and DMSP uptake sites in two species of marine bacteria, Appl. Environ. Microb., 63, 3182–3188, 1997.

Zhang, H. H., Yang, G. P., and Zhu, T.: Distribution and cycling of dimethylsulfide (DMS) and dimethylsulfoniopropionate (DMSP) in the sea-surface microlayer of the Yellow Sea, China, in spring, Cont. Shelf Res., 28, 2417–2427, https://doi.org/10.1016/j.csr.2008.06.003, 2008.

Zindler-Schlundt, C., Lutterbeck, H., Endres, S., and Bange, H. W.: Environmental control of dimethylsulfoxide (DMSO) cycling under ocean acidification, Environ. Chem., 13, 330–339, https://doi.org/10.1071/EN14270, 2015.

[Figure]

2        **Figure 1.** Location of the sampling station (123.50 °E, 30.56 °N) during the cruise.

[Figure]

3  **Figure 2.** Temporal changes in pH (a) for averages of the nine barrels over the triplicates treatments during ocean acidification incubation

4     experiment, variations of light intensity (b) during photodegradation experiment and temperature (c) during temperature experiment.

[Figure]

2 **Figure 3.** Temporal changes in the Chl-*a* concentrations (μg L$^{-1}$) for (a) each of the nine barrels and (b) averages over the

3 triplicates treatments.

[Figure]

2    **Figure 4.** Temporal changes in the DMS concentrations (nmol L$^{-1}$) for (a) each of the nine barrels and (b) averages over the

3    triplicates treatments.

[Figure]

3 **Figure 5.** Temporal changes in the DMSPd and DMSPp concentrations (nmol L$^{-1}$) for (a, c) each of the nine barrels and (b, d)

4 averages over the triplicates treatments, respectively.

[Figure]

3 **Figure 6.** Temporal changes in the DMSOd and DMSOp concentrations (nmol L$^{-1}$) for (a, c) each of the nine barrels and (b,

4 d) averages over the triplicates treatments, respectively.

[Figure]

2 **Figure 7.** Average values of (a) DMS and (b) DMSPp concentrations in the cross experiment of OA and temperature, with

3 three replicate incubation samples per treatment.

[Figure]

2 **Figure 8.** Relationships between bacterial abundance, (a) DMS and (b) DMSOd concentrations in the LC (pH 8.1, filled

3 circle, solid lines), MC (pH 7.9, empty triangle, dash lines) and HC (pH 7.7, empty circular, dotted lines) treatments during

4 the experiment.

[Figure]

3  **Figure 9.** Mean ratios of (a) DMS/Chl-*a*, (b) DMSPt/Chl-*a*, (c) DMSOt/Chl-*a* for the LC, MC and HC treatments.

1    **Table 1.** Preliminary carbonate parameters of three treatments during the incubation experiment.

| Treatments | pH ±s.d. | DIC ±s.d. ($\mu mol\ kg^{-1}$) | $pCO_2$ ±s.d. ($\mu atm$) | $HCO_3^-$ ±s.d. ($\mu mol\ kg^{-1}$) | $CO_3^{2-}$ ±s.d. ($\mu mol\ kg^{-1}$) |
|---|---|---|---|---|---|
| M1-3 | 8.1 ±0.008 | 2271 ±0.577 | 389.2 ±5.661 | 2087.2 ±3.371 | 168.0 ±3.926 |
| M4-6 | 7.9 ±0.012 | 2263 ±1.000 | 635.9 ±13.80 | 2130.0 ±4.133 | 107.1 ±4.093 |
| M7-9 | 7.7 ±0.032 | 2274 ±1.154 | 1040.7 ±54.05 | 2164.0 ±4.193 | 67.5 ±7.552 |

1 **Table 2.** The mean dimethylated sulfur compounds and Chl-*a* concentrations and bacterial abundance in the different

2 treatments over the entire experiment.

| Treatments (pH $\pm$ s.d.) | Sample concentrations | | |
| --- | --- | --- | --- |
| | Sample | Minimum | Maximum | Mean $\pm$ s.d. |
| M1-3 | Chl-*a* ($\mu$g L$^{-1}$) | 0.06 | 12.6 | 3.14 $\pm$ 3.22 |
| (pH = 8.11 $\pm$ 0.017) | Bacterial abundance ($10^8$ cell L$^{-1}$) | 0.29 | 5.06 | 2.09 $\pm$ 1.78 |
| | DMS (nmol L$^{-1}$) | 3.03 | 55.8 | 16.5 $\pm$ 12.4 |
| | DMSPd (nmol L$^{-1}$) | 3.54 | 71.2 | 20.2 $\pm$ 12.5 |
| | DMSPp (nmol L$^{-1}$) | 17.9 | 132 | 69.5 $\pm$ 31.9 |
| | DMSOd (nmol L$^{-1}$) | 26.9 | 85.8 | 44.9 $\pm$ 11.7 |
| | DMSOp (nmol L$^{-1}$) | 6.99 | 64.9 | 32.3 $\pm$ 12.8 |
| M4-6 | Chl-*a* ($\mu$g L$^{-1}$) | 0.08 | 11.1 | 2.60 $\pm$ 2.31 |
| (pH = 7.90 $\pm$ 0.022) | Bacterial abundance ($10^8$ cell L$^{-1}$) | 0.23 | 6.07 | 2.78 $\pm$ 2.12 |
| | DMS (nmol L$^{-1}$) | 1.79 | 45.5 | 13.4 $\pm$ 12.0 |
| | DMSPd (nmol L$^{-1}$) | 2.88 | 56.5 | 18.9 $\pm$ 11.8 |
| | DMSPp (nmol L$^{-1}$) | 12.9 | 148 | 57.5 $\pm$ 27.3 |
| | DMSOd (nmol L$^{-1}$) | 18.3 | 75.8 | 37.9 $\pm$ 6.64 |
| | DMSOp (nmol L$^{-1}$) | 8.24 | 64.0 | 31.4 $\pm$ 11.6 |
| M7-9 | Chl-*a* ($\mu$g L$^{-1}$) | 0.03 | 10.0 | 2.14 $\pm$ 1.72 |
| (pH = 7.72 $\pm$ 0.029) | Bacterial abundance ($10^8$ cell L$^{-1}$) | 0.44 | 9.60 | 3.74 $\pm$ 3.35 |
| | DMS (nmol L$^{-1}$) | 1.98 | 30.6 | 10.7 $\pm$ 7.25 |
| | DMSPd (nmol L$^{-1}$) | 2.93 | 83.2 | 19.0 $\pm$ 12.4 |
| | DMSPp (nmol L$^{-1}$) | 14.4 | 89.3 | 54.6 $\pm$ 21.7 |
| | DMSOd (nmol L$^{-1}$) | 18.5 | 71.7 | 37.9 $\pm$ 10.7 |
| | DMSOp (nmol L$^{-1}$) | 7.67 | 64.2 | 29.4 $\pm$ 11.9 |

1 **Table 3.** DMS photolysis rate constants and turnover time in the cross experiment of OA and light, with three replicate

2 incubation samples per treatment.

| pH | $K_{natural\ light}$ (d$^{-1}$) | $\tau_{photo}$ (d) | $K_{UVB}$ (d$^{-1}$) | UVB contributions | $K_{UVA}$ (d$^{-1}$) | UVA contributions | $K_{visible\ light}$ (d$^{-1}$) | Visible light contributions |
|---|---|---|---|---|---|---|---|---|
| 8.1 | 4.02 | 1.19 | 1.13 | 28.0% | 2.37 | 59.1% | 0.52 | 12.9% |
| 7.9 | 5.19 | 0.92 | 3.20 | 61.7% | 1.53 | 29.5% | 0.46 | 8.80% |
| 7.7 | 6.32 | 0.76 | 4.73 | 74.9% | 1.35 | 21.3% | 0.24 | 3.83% |
| Average | 5.18 | 0.96 | 3.02 | 55.0% | 1.75 | 37.0% | 0.41 | 9.00% |

1    **Table 4.** Pearson's correlation coefficient and associated significance level for mean dimethylated sulfur compounds and

2    Chl-*a* concentrations under three different $CO_2$ treatments (n=15).

| | Chl-*a* | DMS | DMSPd | DMSPp | DMSOd | DMSOp |
|---|---|---|---|---|---|---|
| **LC (pH = 8.1)** | | | | | | |
| Chl-*a* | 1 | | | | | |
| DMS | **.925**[**] | 1 | | | | |
| DMSPd | **.946**\*\* | **.923**\*\* | 1 | | | |
| DMSPp | **.724**\*\* | **.732**\*\* | .570[*] | 1 | | |
| DMSOd | .791[*] | **.778**\*\* | **.789**\*\* | .522[*] | 1 | |
| DMSOp | .334 | .380 | .119 | **.816**\*\* | .200 | 1 |
| **MC (pH = 7.9)** | | | | | | |
| Chl-*a* | 1 | | | | | |
| DMS | **.913**\*\* | 1 | | | | |
| DMSPd | **.769**\*\* | **.784**\*\* | 1 | | | |
| DMSPp | .409 | .445 | .198 | 1 | | |
| DMSOd | .541[*] | .533[*] | .626[*] | .481 | 1 | |
| DMSOp | .400 | .382 | .137 | **.929**\*\* | .545[*] | 1 |
| **HC (pH = 7.7)** | | | | | | |
| Chl-*a* | 1 | | | | | |
| DMS | **.657**\*\* | 1 | | | | |
| DMSPd | **.756**\*\* | **.806**\*\* | 1 | | | |
| DMSPp | .323 | .570[*] | .218 | 1 | | |
| DMSOd | .308 | .238 | .080 | **.807**\*\* | 1 | |
| DMSOp | .265 | .439 | .157 | **.951**\*\* | **.896**\*\* | 1 |

3    \*\*. Correlation is significant at the 0.01 level (2-tailed).

4    \*. Correlation is significant at the 0.05 level (2-tailed).